# Age and Origin of Silicocarbonate Pegmatites of the Adirondack Region

**Jeffrey Chiarenzelli [1,\*], Marian Lupulescu [2], George Robinson [1], David Bailey [3] and Jared Singer [4]**

1   Department of Geology, St. Lawrence University, Canton, NY 13617, USA
2   New York State Museum, Research and Collections, Albany, NY 12230, USA
3   Geosciences Department, Hamilton College, Clinton, NY 13323, USA
4   Earth and Environmental Sciences, Rensselaer Polytechnic Institute, Rensselaer, NY 12180, USA
*   Correspondence: jchiaren@stlawu.edu; Tel.: +1-315-229-5202

**Abstract:** Silicocarbonate pegmatites from the southern Grenville Province have provided exceptionally large crystal specimens for more than a century. Their mineral parageneses include euhedral calc–silicate minerals such as amphibole, clinopyroxene, and scapolite within a calcite matrix. Crystals can reach a meter or more in long dimension. Minor and locally abundant phases reflect local bedrock compositions and include albite, apatite, perthitic microcline, phlogopite, zircon, tourmaline, titanite, danburite, uraninite, sulfides, and many other minerals. Across the Adirondack Region, individual exposures are of limited aerial extent (<10,000 m$^2$), crosscut metasedimentary rocks, especially calc–silicate gneisses and marbles, are undeformed and are spatially and temporally associated with granitic pegmatites. Zircon U–Pb results include both Shawinigan (circa 1165 Ma) and Ottawan (circa 1050 Ma) intrusion ages, separated by the Carthage-Colton shear zone. Those of Shawinigan age (Lowlands) correspond with the timing of voluminous A-type granitic magmatism, whereas Ottawan ages (Highlands) are temporally related to orogenic collapse, voluminous leucogranite and granitic pegmatite intrusion, iron and garnet ore development, and pervasive localized hydrothermal alteration. Inherited zircon, where present, reflects the broad range of igneous and detrital ages of surrounding rocks. Carbon and oxygen isotopic ratios from calcite plot within a restricted field away from igneous carbonatite values to those of typical sedimentary carbonates and local marbles. Collectively, these exposures represent a continuum between vein-dyke and skarn occurrences involving the anatexis of metasedimentary country rocks. Those of Ottawan age can be tied to movement and fluid flow along structures accommodating orogenic collapse, particularly the Carthage-Colton shear zone.

**Keywords:** silicocarbonate; Grenville Province; Adirondack Mountains; Carthage-Colton shear zone; U–Pb zircon geochronology; exhumation; Ottawan Orogeny; calc–silicate minerals; vein-dykes; skarns

## 1. Introduction and Geological Setting

The southern Grenville Province (Figure 1) contains important mineral localities that have produced exceptional crystal specimens that now grace exhibits in many natural history and mineral museums, as well as private collections worldwide. A unique subset of these localities, termed "vein-dykes" [1], are primarily located in the Central Metasedimentary Belt of Ontario and Quebec, with similar occurrences in the Adirondack Mountains of northern New York (Figure 1). Whereas these localities are well-known to collectors and mineral enthusiasts, they have received relatively little attention from geologists despite the potential constraints they offer on the tectonic evolution of the region [2,3]. These occurrences are characterized by their intrusive nature and abundance of large, well-formed crystals of calc–silicate minerals, including pyroxenes, amphiboles, and scapolite, and

other trace minerals such as albite, apatite, zircon, titanite, K-feldspar (primarily perthitic microcline), quartz, tourmaline, and mica, and various sulfide minerals, within a calcite matrix. Weathering leading to dissolution of calcite facilitates the discovery of crystal "pockets" and the removal of intact calc–silicate crystals. Numerous origins have been proposed for the vein-dykes including as skarns, anatexis of marble/calc–silicate country rock, carbonatites, and silicocarbonate melts of crustal or mantle origin, and are summarized in Joyce [4].

Much of the previous published work on vein-dykes addresses those in the Central Metasedimentary Belt and its boundary zones located in Ontario and Quebec. Ellsworth [1] interpreted them as pegmatites and first used the term vein-dyke in reference to their spatial relations with the country rock. Moyd [5] interpreted em as remobilized marbles and attributed the calc–silicate minerals they contain to components of the lithologies encountered during melting, transport, and intrusion. Lumbers et al. [2] considered them pegmatites and carbonatites associated with wide, linear zones of fenitization. Lentz [6] suggested they formed by the melting of marble intruded by granitic pegmatite, with melting assisted by fluxing agents, and suggested they were carbonatitic melts [7]. Based on numerous lines of geochemical and isotopic evidence, Moecher et al. [3] considered those in the Central Metasedimentary Belt boundary zone metamorphosed carbonatites. Sinaei-Esfahani [8] noted features attributed to melting, oxygen and carbon isotopes in calcite as intermediate between crustal and mantle sources, and applied the term silicocarbonates to rocks exposed along Autoroute 5 in Old Chelsea, Quebec. Schumann et al. [9] presented evidence for silicocarbonate melt inclusions in apatite from Otter Lake, Quebec. Robinson et al. [10] noted the linear arrangement of similar occurrences in the Adirondack Lowlands and their correspondence to belts of carbonate-rich metasedimentary rocks. Herein, we utilized the term "silicocarbonate pegmatite" to recognize the abundance of calc–silicate minerals and the pegmatitic grain size they have in the study area.

In this study, we examined silicocarbonate pegmatitic rocks in the Adirondack Region across a structural boundary known as the Carthage-Colton shear zone (CCsz, [11]). This shear zone forms the boundary between the Adirondack Highlands and Lowlands (Figure 1). The Adirondack Lowlands are dominated by supracrustal rocks of the Grenville Supergroup [12], last deformed and metamorphosed to upper amphibolite facies during the Shawinigan Orogeny (circa 1200–1140 Ma). In contrast, the granulite-facies Highlands are dominated by metaigneous rocks of the anorthosite–mangerite–charnockite–granite (AMCG) suite and were deformed and metamorphosed during both the Shawinigan and Ottawan (circa 1090–1020 Ma) events [13]. We present first-order data including field relations, U–Pb zircon ages, zircon rare-earth element patterns, and carbon and oxygen isotope ratios from calcite to constrain their origin and significance to the region's geological history. We find both Shawinigan and Ottawan crystallization ages, generally separated by the Highlands-Lowlands boundary (CCsz; Figure 2), and conclude that the occurrences represent a continuum between classic skarn mineralization and vein-dykes. Those temporally associated with the end of the Shawinigan Orogeny formed along the contact of large syenite–granite bodies and marble, and generally contain wollastonite. Those formed during the Ottawan Orogeny are silicocarbonate melts generated by anatexis of metasedimentary rocks generated during orogenic collapse and exhumation, associated with major faults and lithologic boundaries that served as foci for fluids.

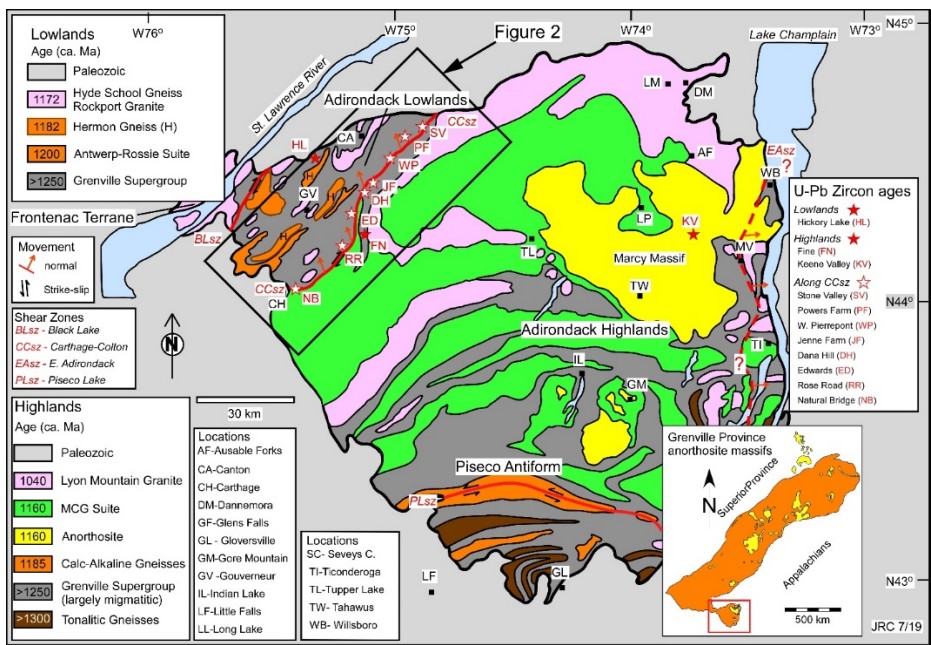

**Figure 1.** Simplified geologic map of the Adirondack Region. Inset shows the location of major anorthosite bodies in yellow within the contiguous Grenville Province (in orange) and the area covered by the figure (red rectangle). The black, diagonally oriented rectangle is shown in additional detail in Figure 2. Stars represent sampling sites described in the text. Figure modified from Chiarenzelli et al. [14].

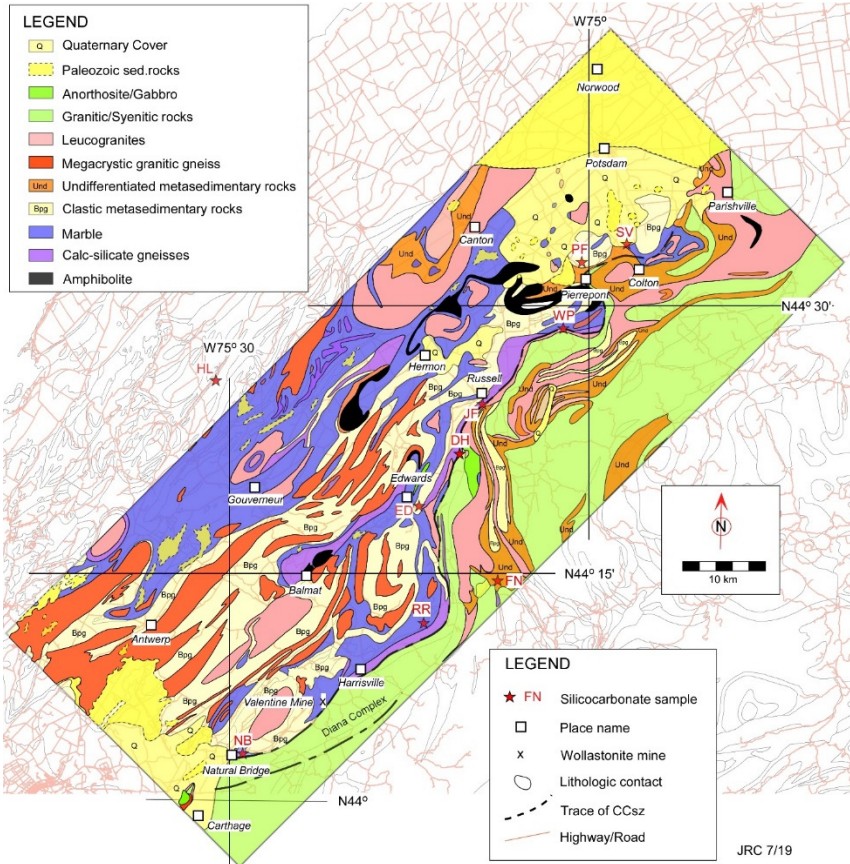

**Figure 2.** Simplified geological map of the boundary between the Adirondack Highlands and Lowlands after Isachsen and Fisher [11]. Red stars represent sample localities along the Carthage-Colton shear zone.

## 2. Samples and Analytical Methods

### 2.1. Field Relations of Samples

Table 1 lists the location, dominant minerals, and age of each of the silicocarbonate pegmatites sampled for geochronology. The samples were collected from silicocarbonate pegmatite exposures in the Adirondack Highlands and Lowlands and from within the Carthage-Colton shear zone. The most notable features of these mineral occurrences are their size and the euhedral shape of calc–silicate minerals. For example, tremolite or edenite crystals at the Jenne Farm (JF) range up to a half-meter in length, whereas crystals of scapolite from the Keene Valley airport (KV) reach several meters in length [15]. In general, most occurrences are of limited areal extent (exposures of a few 100 m to 1000s of m$^2$), some, such as the West Pierrepont Selleck Road tremolite occurrence, however, are exposed as narrow lithologically bound zones with nearly continuous to sporadic development over several kilometers or more, linking several historic collecting sites. In most, an intrusive origin is clear, with veins containing euhedral crystals in a calcite matrix or pockets, cross-cutting the foliation and compositional layers within host lithologies.

**Table 1.** U–Pb zircon sample locations, dominant mineralogy, and U–Pb zircon age.

| Sample Name[b] (Abbreviation) | Lat/Long Decimal ° | Dominant Minerals | Interpreted Age (Ma) |
|---|---|---|---|
| Lowlands | | | |
| Hickory Lake (HL) | 44.465–75.493° | Tur–Ab–Scp–Bt–Dan | 1168 ± 2.8 |
| Highlands | | | |
| Fine (FN) | 44.247–75.135° | Kfs–Cal–Hd–Ap | 1049 ± 4.6 |
| Keene Valley (KV) | 44.222–73.791° | Scp–Hd–Cal–Ttn | 1025 ± 1.7 |
| Along Carthage-Colton shear zone (SW to NE) | | | |
| Natural Bridge (NB) | 44.074–75.473° | Tr–Di–Kfs–Ttn–Wo–Cal | 1168 ± 7.6 |
| Rose Road (RR) | 44.200–75.233° | Ab–Di–Ttn–Wo–Ap | 1163 ± 20 |
| Edwards (ED) | 44.316–75.240° | Di–Cal–Bt–Phl | 1056 ± 3.7 |
| Dana Hill (DH) | 44.369–75.187° | Di–Dan–Cal–Qtz | 1040 ± 5.0 |
| Jenne Farm (JH) | 44.418–75.155° | Tr–Cal | 1060 ± 19 |
| W. Pierrepont (WP) | 44.491–75.037° | Tr–Di–Cal–Tur–Bt–Scp | 1151 ± 6 |
| Powers Farm (PF) | 44.556–75.021° | Di–Act–Phl–Qtz–Cal–Tur | 1152 ± 3.9 |
| Stone Valley (SV) | 44.573–74.950° | Di–Cal–Ksp | 1041 ± 4.4 |

Abbreviations: Ab—Albite; Act-Actinolite; Ap—Apatite; Bt—Biotite; Cal—Calcite; Dan—Danburite; Di—Diopside; Hd—Hedenbergite; Kfs—Potassium Feldspar; Phl—Phlogopite; Qtz—Quartz; Scp—Scapolite; Ttn—titanite; Tur—Tourmaline; Tr—Tremolite; Wo—Wollastonite.

Intrusion and cross-cutting relations are particularly well demonstrated at Powers Farm (PF), in Pierrepont, NY, where a set of vertical, parallel veins extending for over 20 m crosscut a strong foliation developed in a compositionally variable package of metasedimentary rocks of the Grenville Supergroup [16]. In addition, at Stone Valley, along the trace of the Carthage-Colton shear zone near Colton, NY, silicocarbonate pegmatite fills the interstices between meter-scale, mylonitic blocks in a mega-breccia. At some localities, xenoliths of country rock up to a meter or more in diameter are found. For example, at Edwards (ED), a diopside-rich marble xenolith is entrained in the silicocarbonate. Occurrences are largely associated with calc–silicate, marble, and related metasedimentary lithologies that form a major part of the structurally complex stratigraphic sequence exposed in the Lowlands. At most localities, granitic pegmatites are located nearby and indicate the contemporaneous intrusion of felsic melts. The minerals and proportions of minerals at each locality, and among localities, vary considerably and generally reflect the immediately adjacent bedrock.

## 2.2. U–Pb Zircon Geochronology and Trace-Element Analysis

The authors collected the silicocarbonate pegmatite samples from natural outcroppings or road cut exposures (Table 1). Sampling localities included known mineral sites on public property or where access to private property and sample collection were allowed or permitted for a fee. The samples included examples of silicocarbonate pegmatites from both the Adirondack Highlands and Lowlands terranes, and along the trace of the Carthage-Colton shear zone. Samples ranging from 1–2 km were collected from areas of outcrops with moderate grain size, with representative minerals, generally including larger calc–silicate crystals.

Samples for geochronology were processed, after initial crushing, at the Arizona Laserchron Center using standard techniques for heavy-mineral separation. Once separated, the zircon crystals were hand-picked and mounted in epoxy plugs for analysis and imaging. A Hitachi 3400N scanning electron microscope (SEM) (Hitachi Ltd., Tokyo, Japan) was operated in back-scattered electron (BSE) or cathodoluminescence (CL) modes to document each zircon population prior to analysis. The SEM images were used to target specific zircon grains and areas on each grain for laser ablation (30 μm diameter pits). After analysis, a subset of zircons from each sample was selected for trace-element analysis conducted at Rensselaer Polytechnic Institute by LA-ICP-MS (Photon Machines Analyte 193 G1 short-pulse eximer laser coupled to a Varian 820 quadrapole ICP-MS, Isomass Scientific Inc., Calgary, AB, Canada). Complete analytical details are given in Appendix A.

Zircon crystals are typically found included in silicate phases (i.e., pyroxenes, amphiboles, micas, feldspar) within these rocks. For example, coauthor ML first noted the occurrence of zircon in tourmaline–quartz intergrowths from Powers Farm. In other rocks, zircon occurs as dispersed crystals of near equal size to the other minerals in the rock, apparently having grown, more or less, synchronously with other phases. In some samples, the exact location of the zircon crystals is unknown as they were not in great abundance or large size and were found during the separation process.

## 2.3. Carbon and Oxygen Isotopic Analysis of Calcite

Samples of calcite were removed from silicocarbonate samples using a rock hammer and were hand purified under binocular microscope (EZ4 stereo microscope, Leica Microsystems, Wetzlar, Germany) examination to remove other minerals and submitted for analysis. At several sampling localities, multiple samples were collected to test reproducibility. Calcite samples were prepared for C and O isotope analysis by powdering in an agate mortar at Hamilton University in Clinton, NY. Between 120–160 μg of powdered sample were loaded into autosampler vials and analyzed on a Thermo Delta V (Delta V continuous flow isotope ratio mass spectrometer, Thermo Scientific, Bremen, Germany) and Gasbench (Gasbench II, Thermo Finnegan, Bremen, Germany) instrument. All analytical runs were bracketed by laboratory standards (LECO, NBS-18, $CaCO_3$-Merk, NBS-19), calibrated to the Vienna Pee Dee Belemnite (V-PDB) and Vienna Standard Mean Ocean Water (VSMOW) reference standards.

## 3. Results

The location of each sample collection site is shown in Figure 2. Photographs of some field locations are found in Figure 3. Representative optical and scanning electron images of zircon are shown in Figure 4. The results of U–Pb zircon analyses, REE in zircon, and carbon and oxygen isotopes of calcite are found, respectively, in Figures 5–7. Tables 1 and 2 summarize the results of U–Pb zircon analyses. The complete U–Pb zircon dataset is available in Supplemental File S1. Analytical procedures are found in Appendix A. The mean square weighted deviation (MWSD) are listed for each age determined where applicable.

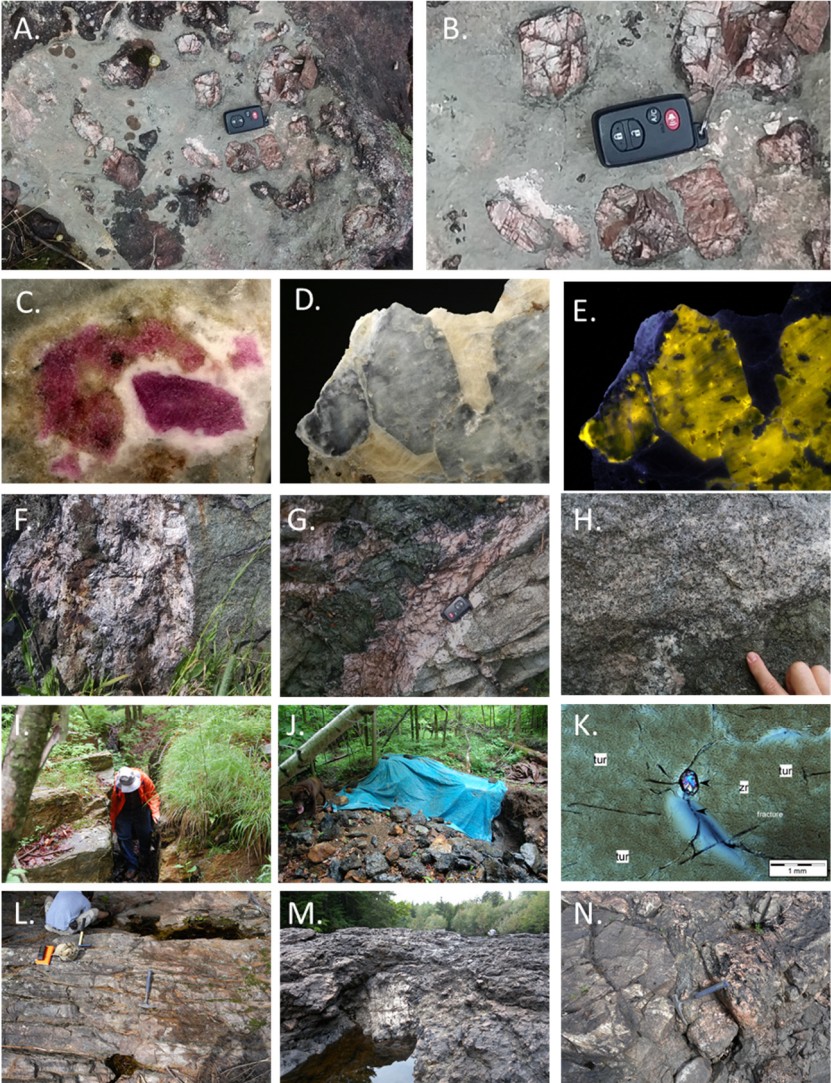

**Figure 3.** Photographs of select sampling sites. (**A**) Perthitic microcline crystals in calcite matrix, Fine locality. Key fob was 5 cm long. (**B**) Closeup of (**A**) showing euhedral, perthitic microcline crystals showing cleavage surfaces. (**C**) Corundum from the purple diopside mound, Rose Road locality (field of view 2 cm). (**D**) Scapolite crystals in calcite matrix, Rose Road locality (field of view 10 cm). (**E**) Same shot as in 3D viewed in long wave ultraviolet light displaying lemon yellow fluorescence of scapolite and dark blue fluorescence in calcite. (**F**) Contact between pink calcite pod and granular diopsidite, Edwards locality. The iron staining is from the oxidation of sulfide minerals, primarily chalcopyrite. The horizontal width of calcite pod was 1.5 m. (**G**) Pink calcite vein with large, euhedral diopside crystals (left) and granular diopsidite (right) from Edwards. Key fob for scale. (**H**) Diopside-rich marble xenolith in Edwards outcrop. (**I**) One of five parallel, north-trending trenches at Powers Farm. Coauthor George Robinson for scale. (**J**) Tarp (blue) covered trench (length = 2.44 m) at Powers Farm at one of the numerous dig sites. Note the tourmaline-rich debris (black) in the foreground to the right of Crunch, the chocolate lab. (**K**) Photomicrograph showing 500 μm long zircon crystal in tourmaline from Powers Farm, cross polarizers (tur—tourmaline; zr—zircon). (**L**) Closely spaced brittle fractures cutting early foliation at Stone Valley. Fractures accentuatedj by calcite dissolution. (**M**) Large, meter-scale mylonitic block in tectonic breccia at Stone Valley in the bed of the Raquette River. Ben Valentino for scale (upper right). (**N**) Pink pegmatitic pod between mylonitic breccia fragments, Stone Valley. Hammer for scale.

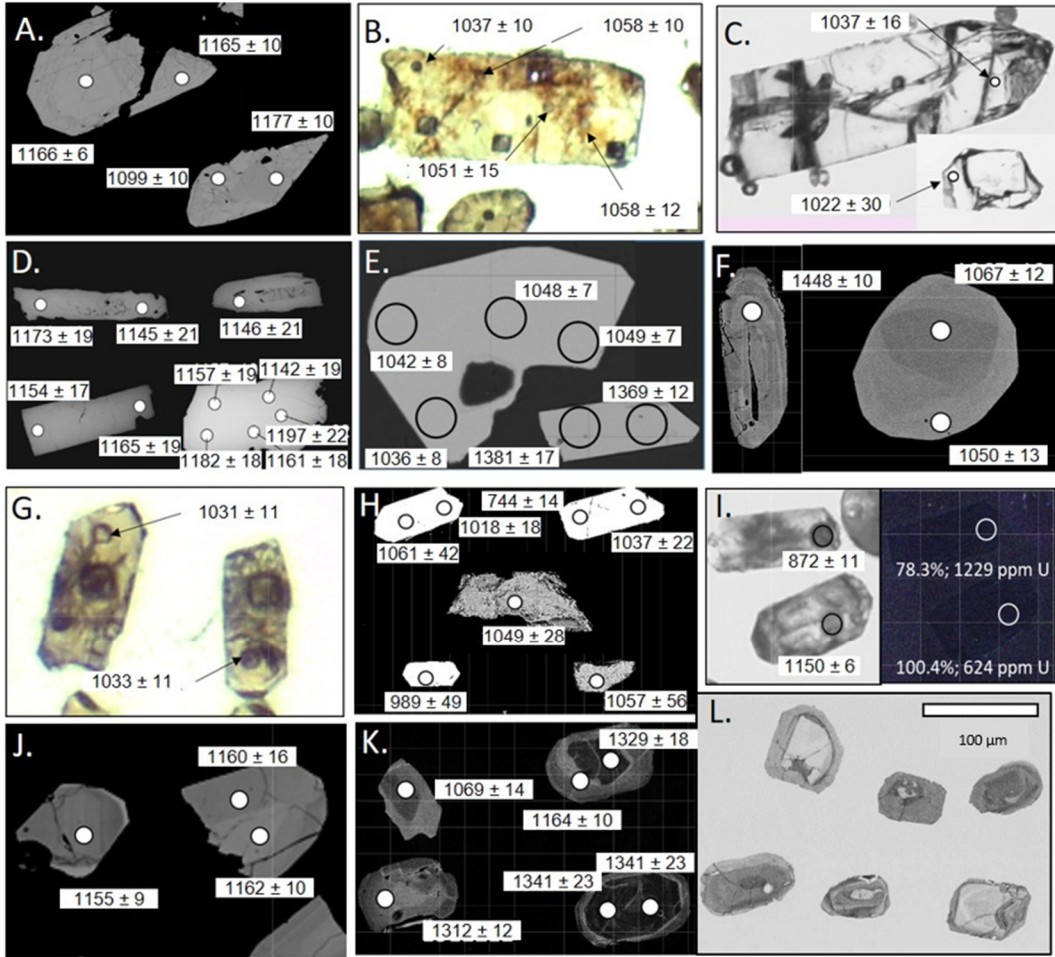

**Figure 4.** Representative photomicrographs and scanning electron microscope images of the zircon crystals analyzed during this study. All circular ablation pits were 30 μm in diameter; ages are given with an uncertainty in Ma. (**A**) Hickory Lake locality. Note the younger age in the outer rim of the grain in the SE corner, BSE. (**B**) Fine locality. Large zircon analyzed several times, polarized light. (**C**) Keene Airport. Representative crystals, Note the large size and euhedral shape, polarized light. (**D**) Natural Bridge locality. Note the elongate grains, each with a fractured core (upper two grains), BSE. (**E**) Rose Road locality. Note the differences in size, shape, and age, BSE. (**F**) Edwards locality. Note the distinct shapes and ages of the crystals, cathodoluminscence (CL). (**G**) Dana Hill locality. Note the dark coloration of the grains and circular U–Pb and square (trace-element) ablation pits, polarized light. (**H**) Jenne Farm locality. Note the difference in shape and response of the crystals to BSE. (**I**) West Pierrepont locality. Left side: crystals in polarized light; right side: zircons in CL. Note the lack of CL response. (**J**) Powers Farm locality. Faceted, but near equant crystals, BSE. (**K**) Stone Valley locality. Note the differences in CL signature and zircon core and rim regions. (**L**) Stone Valley locality. Note the core and rim relations in BSE (inverted). Bar for scale.

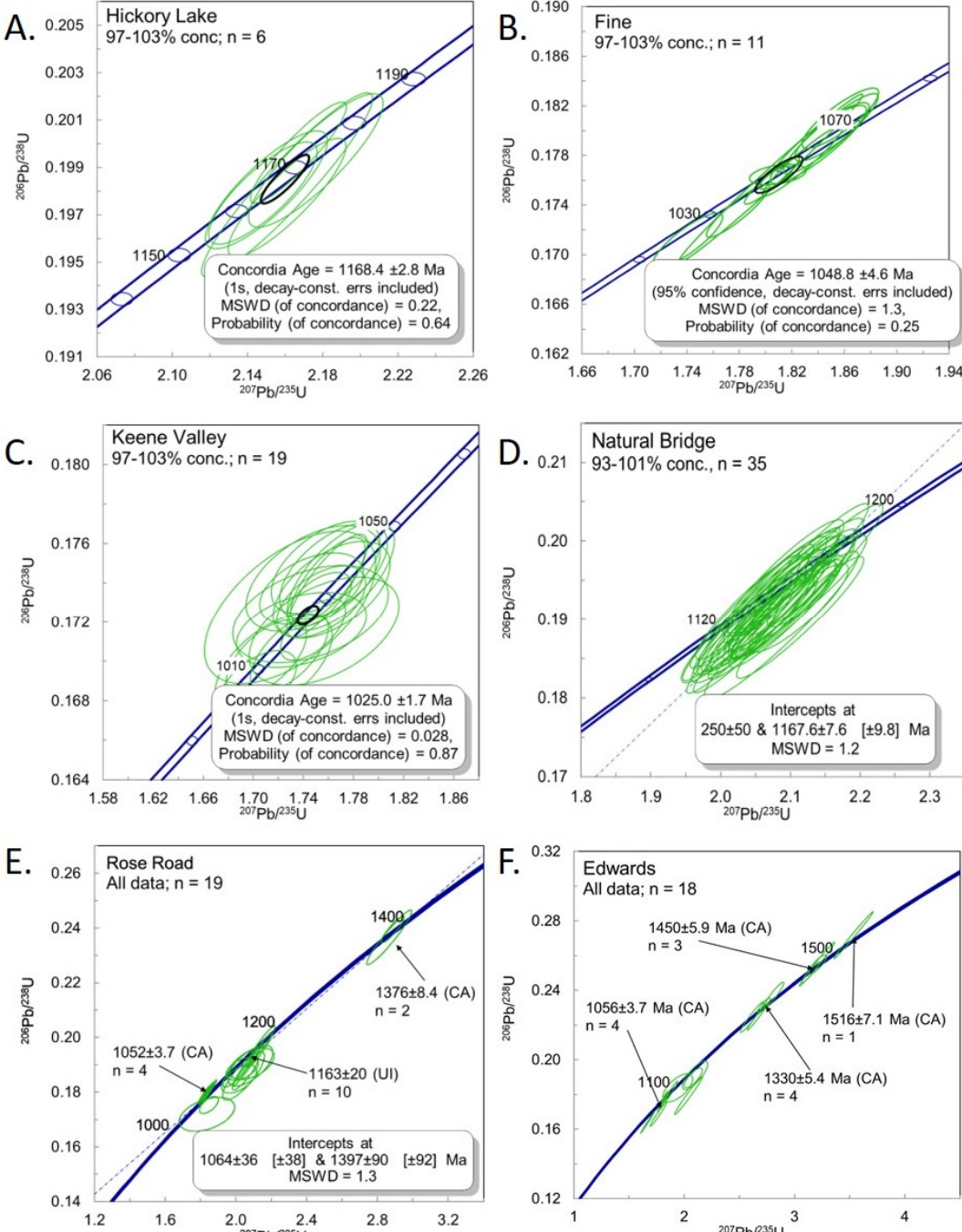

**Figure 5.** *Cont.*

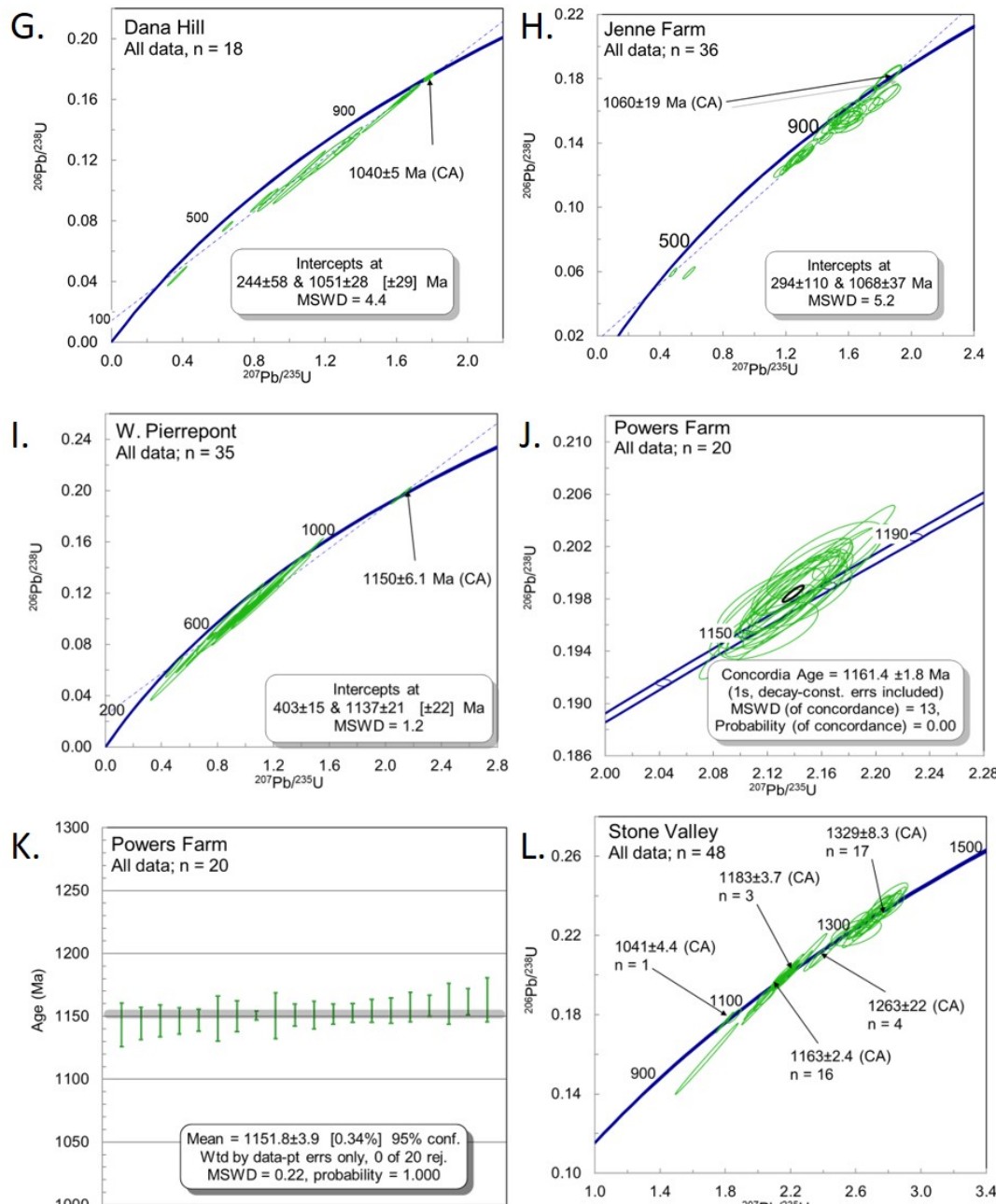

**Figure 5.** Concordia diagrams for the samples analyzed in this study. See text for explanation. Locations include: (**A**) Hickory Lake; (**B**) Fine; (**C**) Keene Valley; (**D**) Natural Bridge; (**E**) Rose Road; (**F**) Edwards; (**G**) Dana Hill; (**H**) Jenne Farm; (**I**) W. Pierrepont; (**J**) Powers Farm; (**K**) Powers Farm (Weighted Mean); (**L**) Stone Valley.

**Table 2.** U–Pb zircon geochronological results from silicocarbonate pegmatites.

| Sample | Number (Comment) | Age (Ma) [1] | MSWD [2] | U (ppm) | U/Th Ratio |
|---|---|---|---|---|---|
| Dana Hill (DH) | 18 (all) | 1051 ± 28 UI | 4.4 | 1128 ± 368 | 13.1 ± 3.6 |
| | 1 (99.6%) | 1040 ± 5 CA | 0.13 | 879 | 9.5 |
| | 6 (>90 conc.) | 1035 ± 5.9 WA | 0.53 | 876 ± 34 | 9.5 ± 0.5 |
| Edwards (ED) | 4 (group) | 1056 ± 3.7 CA/1055 ± 7.7 WA | 0.35/1.05 | 444 ± 223 | 33.5 ± 7.6 |
| | 4 (group) | 1330 ± 5.4 CA/1328 ± 23 WA | 0.51/1.6 | 242 ± 99 | 1.3 ± 0.2 |
| | 3 (group) | 1450 ± 5.9 CA/1449 ± 12 WA | 0.13/0.03 | 179 ± 32 | 1.6 ± 0.2 |
| | 1 (102.3%) | 1516 ± 7.1 CA | 0.56 | 479 | 1.9 |
| Fine (FN) | 15 (all) | 1054 ± 5.8 UI | 0.34 | 814 ± 119 | 14.6 ± 1.3 |
| | 11(>97% conc.) | 1049 ± 4.6 CA/1053 ± 4.6 WA | 1.3/0.49 | 790 ± 115 | 15.1 ± 1.3 |
| Hickory Lake (HL) | 12 (all) | 1153 ± 22 UI | 7.0 | 675 ± 644 | 9.22 ± 6.4 |
| | 6 (group) | 1168 ± 2.8 CA/1170 ± 7.3 WA | 0.22/0.47 | 425 ± 85 | 5.9 ± 1.6 |
| | 3 (group) | 1097 ± 13 CA/1110 ± 31 WA | 6.8/1.9 | 364 ± 132 | 16.6 ± 4.7 |
| Keene Valley (KV) | 19 | 1025 ± 1.7 CA/1024 ± 10 WA | 0.03/0.15 | 156 ± 35 | 4.6 ± 1.1 |
| | 1(94.5% conc.) | 1103 ± 64 7/6 | NA | 170 | 3.8 |
| Jenne Farm (JF) | 36 | 1068 ± 37 UI | 5.2 | 828 ± 722 | 19.6 ± 15.2 |
| | 1 (102.1%) | 1060 ± 19 CA | 0.32 | 1397 | 24.7 |
| | 1 (82.9%) | 1623 ± 27.2 7/6 | NA | 532 | 6.6 |
| Natural Bridge (NB) | 35 (all) | 1168 ± 7.6 UI | 1.2 | 680 ± 69 | 7.0 ± 1.1 |
| | 35 (all) | 1146 ± 2.8 CA/1162 ± 6.9 WA | 30.0/1.2 | 680 ± 69 | 7.0 ± 1.1 |
| Powers Farm (PF) | 20 (all) | 1152 ± 9.4 UI | 0.23 | 313 ± 115 | 3.5 ± 0.8 |
| | 20 (all) | 1161 ± 1.8 CA/1152 ± 3.9 WA | 13/0.22 | 313 ± 115 | 3.5 ± 0.8 |
| Rose Road (RR) | 4 (group) | 1052 ± 3.7 CA/1045 ± 7.3 WA | 6.0/0.68 | 474 ± 28 | 33.6 ± 0.5 |
| | 10 (group) | 1163 ± 20 UI/1172 ± 10 WA | 0.24/0.25 | 121 ± 93 | 1.5 ± 0.8 |
| | 2 (group) | 1376 ± 8.4 CA/1373 ± 19 WA | 0.19/0.29 | 98 ± 5 | 1.1 ± 0.1 |
| Stone Valley (SV) | 1 (101.5%) | 1041 ± 4.4 CA | 0.07 | 592 | 39.8 |
| | 1 (99.4%) | 1066 ± 9.5 CA | 0.11 | 205 | 44.7 |
| | 16 (group) | 1163 ± 2.4 CA/1161 ± 3.8 WA | 0.83/0.38 | 360 ± 76 | 24.2 ± 8.3 |
| | 3 (group) | 1183 ± 3.7 CA/1187 ± 6.7 WA | 2.5/0.77 | 397 ± 92 | 14.9 ± 11.1 |
| | 4 (group) | 1263 ± 22 CA/1262 ± 12 WA | 0.29/1.16 | 227 ± 114 | 2.6 ± 1.4 |
| | 17 (group) | 1329 ± 8.3 CA/1332 ± 8.7 WA | 0.62/1.3 | 134 ± 84 | 1.7 ± 0.6 |
| W. Pierrepont (WP) | 35 (all) | 1137 ± 21 UI | 1.2 | 1508 ± 582 | 15.6 ± 6.1 |
| | 1 (100.4%) | 1150 ± 6.1 CA | 0.03 | 624 | 4.5 |

[1] Regression technique: CA: concordia age; WA: weighted average; UI: upper intercept; 7/6: $^{207}Pb/^{206}Pb$ age; (100.4%): conc.: % concordancy; NA: not applicable. Group: age cluster. [2] Mean square weighted deviation.

### 3.1. U–Pb Zircon Geochronology

#### 3.1.1. Lowlands Geochronological Samples

Hickory Lake (HL). A sample was analyzed from Hickory Lake approximately 30 km northeast of the CCsz. The Hickory Lake locality is associated with tourmaline (dark brown uvite and, less commonly, blue dravite), biotite, albite, diopside, scapolite, and numerous other minerals. This locality is known as the Downing Farm; Brown [17] first reported minerals of interest. At this locality, the second discovery of danburite in the Adirondacks occurs along the mineralized contact between an albite–quartz pegmatite and marble [18].

About a dozen zircon grains ranging from 200 to 750 µm were mounted for analysis. Back-scattered electron SEM images show significant internal zonation, inclusions, and cross-cutting bright areas attributed to later alteration or metamictization (Figure 4a). Twelve spots were ablated on seven grains. From these twelve analyses, a poorly constrained concordia upper intercept of 1153 ± 22 Ma (MSWD = 7.0) was calculated. A closer examination of the data revealed two concordant clusters. The oldest gave a concordia age of 1168 ± 2.8 (Figure 5a; MSWD = 0.22) and a weighted average

of 1170 ± 7.3 Ma (MSWD = 0.47) with 425 ± 85 ppm U and a U/Th ratio of 5.9 ± 1.6. The younger gave a concordia age of 1097 ± 13 Ma (MSWD = 6.8) and a weighted average of 1110 ± 31 Ma (MSWD = 1.9). All three of these analyses came from "bright" areas of BSE images (U = 364 ± 132 ppm; U/Th = 16.6 ± 4.7). Figure 4a shows that the younger concordant grouping was obtained from the edge of older crystals.

The time of crystallization of the silicocarbonate pegmatite was interpreted as 1168 ± 2.8 Ma. The younger "grouping" of concordant zircon crystals was interpreted, based on their BSE response, U/Th ratios, and age, as the outer rim of the zircon crystals partial reset by post-crystallization processes, perhaps associated with the Ottawan Orogeny or an earlier event at circa 1100 Ma.

### 3.1.2. Highlands Geochronological Samples

Fine (FN): The Fine sample was collected at the intersection of Route (Rt). 58 and Route 3. Here a pegmatitic silicocarbonate rock composed mostly of clinopyroxene and microcline (Figure 3A,B) in a calcite matrix was exposed along Rt. 3 and in small, ledge-like outcrops just to the north of the highway in the woods.

About fifty euhedral, dipyramidal, elongate grains of zircon (Average *l:w* ratio 4:1) were mounted in epoxy for analysis (Figure 4B). The largest grains were approximately 250 μm in long dimension. Most grains displayed thin oscillatory zones. Analysis of 15 spots yielded a nearly concordant grouping with a few slightly discordant results, that gave an age of 1054 ± 5.8 Ma (MSWD = 0.34). A concordia age of 1049 ± 4.6 Ma (Figure 5B; MSWD = 1.3) was obtained from a group of 11 results that are >97% concordant. The weighted average calculated from the same dataset yielded an age of 1053 ± 4.8 (MSWD = 0.49). Uranium content was moderately high and relatively uniform (790 ± 115 ppm U), and the U/Th ratio averaged 15.1 ± 1.3. The time of the crystallization of the silicocarbonate pegmatite was interpreted as 1049 ± 4.6 Ma.

Keene Valley (KV): Located in the forest west of the Keene Valley airport, large boulders with the assemblage scapolite–hedenbergite–calcite–titanite ± andradite rest on the valley floor. Elongate pods of the same assemblage have been found within massif anorthosite exposed along the cliff face above. Individual crystals of meionite-rich scapolite up to 3 m long are observed in loose boulders [15]. Jaffe et al. [15] reported that the calc–silicate pods are enveloped successively by forsterite–diopside marble and ferromonzonite–ferrosyenite gneiss within gabbroic anorthosite. They also note that the large, pyramidally-terminated scapolite crystals are consistent with growth from a magma.

About four-dozen zircon grains and angular fragments were mounted in epoxy for analysis. The largest grains are about 4 mm in length (Figure 4C). Most grains display thin, oscillatory zoning and a euhedral shape. Many contain irregular and bright areas in BSE imaging filled with small inclusions. Nineteen analyses yielded a concordant age of 1025 ± 1.7 Ma (MSWD = 0.03) and weighted average of 1024 ± 10 Ma (Figure 5C; MSWD = 0.15). One point was 94.5% concordant and yielded a poorly constrained $^{207}Pb/^{206}Pb$ age of 1102 ± 64 Ma, interpreted as an inherited grain or core. Uranium content was low and relatively uniform (156 ± 34 ppm U) and the U/Th ratio averaged 4.6 ± 1.1. The crystallization age of the silicocarbonate pegmatite was interpreted to be 1025 ± 1.7 Ma.

### 3.1.3. Geochronological Samples Collected along the Carthage–Colton Shear Zone (from SW to NE)

Natural Bridge (NB): This site has been known since the 1830s [19] and is located approximately 2 km east of the village of Natural Bridge. The sampling site was just south of Rt. 3, in a series of small outcrops and excavated trenches exposing what appears to be a skarn composed of tremolite–diopside–potassium feldspar–titanite–wollastonite–calcite [20]. The outcrop occurs between the Diana Syenite and a marble belt, about one kilometer north of the mapped trace of the CCsz. The area is 15 km southwest of the Valentine wollastonite mine (Figure 3).

About a dozen highly elongate (average *l:w* ratio 3:1 to 6:1) zircon crystals (Figure 4D) and angular fragments were mounted in epoxy for analysis. The largest grains had a long axis of about 1.0 mm. Aside from a few potential cores, the grains showed a uniform BSE response. Thirty-five

near-concordant and overlapping analyses yielded an upper intercept age of 1168 ± 7.6 (Figure 5D; MSWD = 1.2), with the lower intercept anchored at 250 ± 50 Ma. The same 35 analyses yielded a concordia age of 1146 ± 2.8 (MSWD = 30) and weighted average of 1162 ± 6.9 (MSWD = 1.2). The zircon were homogeneous, had moderate U concentrations (680 ± 69 ppm), and a U/Th ratio of 7.0 ± 1.1. The crystallization age of the silicocarbonate pegmatite was interpreted as 1168 ± 7.6 Ma.

Rose Road (RR): Approximately 5 km northeast of the village of Pitcairn and about 3 km northwest of Rt. 3, a series of marble and calc–silicate outcrops occur south of Rose Road. This site is known as the Mulvaney property or MacDonald Sugar bush [21,22]. The area is about 3 km northwest of the CCsz. Two distinct areas of mineral collection, approximately 100 m apart, occur and are informally known as the purple diopside mound and the wollastonite skarn. Corundum crystals (variety ruby) up to 2 cm in diameter are found in the purple diopside mound (Figure 3C) along with scapolite and calcite fluorescent lemon yellow and dark blue, respectively, in long-wave UV light (Figure 3D,E). The sample investigated here comes from the wollastonite "skarn" [22] that contains albite, diopside, titanite, wollastonite, and fluorapatite, as well as, numerous other accessory minerals.

About a dozen crystals of zircon of different shapes and size (50–300 μm) were separated and mounted (Figure 4E). The grains vary from round, subhedral to slightly elongate, euhedral terminated dipyramids. In BSE images, the zircon crystals ranged from bright to dark. Nineteen analyses yielded a spread of ages. Four analyses yielded a concordia age of 1052 ± 3.7 Ma (Figure 5E; MSWD = 6.0) and weighted average of 1045 ± 7.3 Ma (MSWD = 0.68). Ten analyses yielded an upper intercept age of 1163 ± 20 Ma (MSWD = 0.24) and a weighted average of 1172 ± 10 Ma (MSWD = 0.25). Two additional analyses yielded a concordia age of 1376 ± 8.4 Ma (MSWD = 0.19) and a weighted average of 1373 ± 19 Ma (MSWD = 0.29). Uranium contents varied in zircons (98 ± 5 ppm, U/Th = 1.1 ± 0.1) yielding an age of circa 1376 Ma to 474 ± 28 ppm (U/Th = 33.6 ± 0.5) for those whose age was circa 1052 Ma. Three analyses, two of lower concordancy, yielded intermediate ages of 1105 ± 30 Ma (94.2% conc.), 1131 ± 29 Ma (98.4% conc.), and 1132 ± 100 Ma (90.3% conc.). As these were poorly constrained and of relatively low concordancy; no interpretation was made of their significance.

The crystallization age of the silicocarbonate pegmatite was interpreted to be 1163 ± 20 Ma. Older crystals were interpreted to be xenocrystic, whereas younger crystals (circa 1050 Ma) likely reflect late (Ottawan) growth during movement and fluid flow along the Carthage–Colton shear zone.

Edwards (ED): Approximately 2 km southwest of the village of Edwards on the east side of Rt. 58, a striking pink and green outcrop occurs [21]. This is known as the Morgan Farm or the Irish Green outcrop. The area is about 3 km west of the mapped trace of CCsz and about 0.5 km southeast of a mylonitic syenite gneiss. Pockets and veins of calcite (Figure 3F) with large, euhedral diopside crystals (Figure 3G) occur within a medium-grained, granular diopsidite host (Figure 3F) and were exposed during blasting of the roadcut. A meter-wide block of diopside-rich marble occurs within the diopsidite (Figure 3H). Accessory minerals include chalcopyrite, molybdenite, and mica. Behind the roadcut in the woods, euhedral microcline crystals up to 15 cm were collected.

About three dozen small, roundish to prismatic crystals varying in size from 75 to 150 μm were mounted in epoxy (Figure 4F). In BSE images, the zircon ranged from bright to dark. The BSE imaging showed a dark core enclosed within an oscillatory-zoned rim having a brighter response. Eighteen analyses yielded a wide spread of ages. Four analyses yielded a concordia age of 1056 ± 3.7 Ma (Figure 5F; MWSD = 0.35) and weighted average of 1055 ± 7.7 Ma (MWSD = 1.05). Four analyses yielded a concordia age of 1330 ± 5.4 (MWSD = 0.51) and weighted average of 1328 ± 23 Ma (MWSD = 0.16). Three additional analyses yielded a concordia age of 1450 ± 5.9 Ma (MWSD = 0.13) and weighted average of 1449 ± 12 MA (MWSD = 0.03). A single analysis gave a $^{207}Pb/^{206}Pb$ age of 1516 ± 7.1 Ma (MSWD = 0.56). Uranium contents varied from 179 ± 32 ppm (U/Th = 1.6 ± 0.2) for the zircon yielding an age of circa 1450 Ma to 444 ± 223 ppm (U/Th = 33.5 ± 7.6) for the youngest zircon. The crystallization age of the silicocarbonate pegmatite was interpreted as 1056 ± 3.7 Ma. Analyses ranging in age from circa 1330 to 1515 Ma were interpreted as indicative of xenocrystic zircon.

Dana Hill (DH): Approximately 5 km northeast of the village of Edwards on the south side of Dana Hill Road, the historic Russell danburite locality (known as the old Van Buskirk farm) occurs [23,24]. The site is within a few hundred meters of the trace of the CCsz. The occurrence is hosted in a sequence of variable and layered metasedimentary rocks including danburite–diopside-rich layers with the terminated danburite crystals occurring within calcite-cored pods. It is associated with a pyroxene-bearing granitic pegmatite. In addition to danburite, diopside, tourmaline, mica, pyrite, and quartz, Chamberlain et al. [25] identified additional minerals.

About three dozen euhedral and prismatic (average *l:w* = 4:1) crystals, varying in size from 200 to 300 μm in long dimension, were mounted. Optically, the zircon was dark and iron stained. In BSE images, it was very bright, with few internal features visible. A clearer rim (visible in polarized light) overgrowing a darker core was targeted (Figure 4G). Eighteen analyses defined a discordia line with an upper intercept of 1051 ± 28 Ma (MSWD = 4.4) and a lower intercept of 244 ± 58 Ma. A single concordant point anchors the upper intercept and had a $^{207}Pb/^{206}Pb$ age of 1040 ± 5 Ma (MSWD = 0.13; Figure 5G). Uranium contents were high, 1128 ± 368 ppm (U/Th = 13.1 ± 3.6). The crystals were rich in uranium and likely metamict; the crystallization age of the silicocarbonate pegmatite was interpreted as 1040 ± 5 Ma, as given by the age of the concordant analysis.

Jenne Farm (JF): Approximately 1 km south of the village of Russell, in the woods on the west side of County Route 27 Road, tremolite or edenite crystals up to 0.5 m in length occur within the exposed silicocarbonate [26]. The site is within a few hundred meters of the trace of the CCsz. The occurrence is within a sequence of calc–silicate gneisses and variable metasedimentary rocks. It is also associated with a granitic pegmatite.

About 50 euhedral and prismatic (average *l:w* = 3:1) crystals of varying in size from 100 to 300 μm in long dimension were mounted. Back-scattered electron imaging revealed a core- and oscillatory-zoned rim. Some mottling was visible, perhaps indicative of recrystallization or radiation damage (Figure 4H). Thirty-seven analyses were completed and thirty-six defined a discordia line with an upper intercept of 1068 ± 37 Ma (MSWD = 5.2) and a lower intercept of 294 ± 110 Ma. A single concordant point anchored the upper intercept and had a $^{207}Pb/^{206}Pb$ age of 1060 ± 19 Ma (MSWD = 0.32; Figure 5H). Six analyses ranging from 82.2 to 87.0% concordant had $^{207}Pb/^{206}Pb$ ages ranging between 1108 and 1623 Ma and were interpreted as xenocrystic material. Uranium contents were high and variable 1397 ± 532 ppm (U/Th = 19.6 ± 15.2). A subset of crystals that were slightly discordant, had $^{207}Pb/^{206}Pb$ ages of circa 1050 Ma, extremely low uranium contents (approximately 30 ppm), and a U/Th ratio of approximately 2.3. The crystallization age of the silicocarbonate pegmatite was interpreted as 1060 ± 19 Ma, the age of a concordant point. Older ages, up to 1623 Ma, were interpreted as xenocrystic material.

West Pierrepont (WP): Approximately 2 km east of West Pierrepont, a classic tremolite locality is noteworthy for the green color the tremolite found. It is located in the woods approximately 0.5 km south of Selleck Road. Here, the tremolite–diopside–calcite–biotite–scapolite–tourmaline assemblage occurs along a ridge parallel to the CCsz (<100 m to the south) and follows the trend of the bedrock for several kilometers east, eventually bending to the north [16]. The occurrence is within a sequence of calc–silicate gneisses and variable metasedimentary rocks, including marble and quartzite. It is also associated with a pyroxene-bearing granitic pegmatite. Perhaps the best collecting site originally occurred along the lithologic boundary between quartzite and marble; however, cross-cutting mineralized veins also occur in abundance.

About 50 euhedral and prismatic (average *l:w* = 4:1) crystals of zircon varying in size from 100 to 300 μm in long dimension were mounted. A dark core was clearly visible in polarized light and as bright areas in BSE imaging. Consequently, oscillatory-zoned rims were targeted. Although no CL response was apparent (Figure 4I), back-scattered electron imaging revealed a core and an oscillatory-zoned rim. Thirty-five analyses were completed and defined a discordia line with an upper intercept of 1137 ± 21 Ma (MSWD = 1.2) and a lower intercept of 403 ± 15 Ma. A single concordant point anchored the upper intercept and had an age of 1151 ± 6 Ma (MSWD = 0.03; Figure 5I). Uranium contents were high and variable 1508 ± 582ppm (U/Th = 15.6 ± 6.1). Despite the high uranium concentrations and

possible associated Pb loss, the crystallization age of the silicocarbonate pegmatite was interpreted as 1151 ± 6 Ma, based on a single concordant point.

Powers Farm (PF): Approximately 2 km north of the village of Pierrepont, the Powers Farm locality has been producing quality specimens of black tourmaline crystals for more than 100 years [16]. It is one of the largest sites of mineralization, exposed over 100 acres or more. Hosted within a sequence of metasedimentary rocks, collecting localities range from tourmaline–quartz dikes cross-cutting bedrock (Figure 3I) to diopside–actinolite(after diopside)–calcite–mica–tourmaline pits dug by collectors (Figure 3J). The occurrence is about 1 km north of the trace of the CCsz. Recent excavation on private property has found the continuation of north-trending tourmaline veins across Leonard Brook, extending the area of known mineralization. Zircon crystals were observed in thin section as inclusions in tourmaline (Figure 3K).

About 50 euhedral and prismatic to nearly equant crystals of zircon varying in size from 200 to 400 μm in long dimension were mounted. The crystals had an equant shape, sharp terminations, and a unique BSE response (Figure 3J). Back-scattered imaging showed strong oscillatory zoning, large inclusions, and a distinct, small core. Twenty analyses were completed and yielded an upper intercept age of 1152 ± 9.4 (MSWD = 0.23). The same analyses gave a concordant age of 1161 ± 1.8 Ma (MSWD = 13) and a weighted average of 1152 ± 3.9 Ma (MSWD = 0.22). Because of the supraconcordant nature of the analyses (Figure 5J), the weighted average was considered the more accurate age (Figure 5K; MSWD = 0.22). Uranium contents were moderate and variable, 313 ± 115 ppm (U/Th = 3.5 ± 0.8). The crystallization age of the tourmaline-bearing pegmatite was interpreted as 1152 ± 3.9 Ma.

### 3.1.4. Geochronological Samples Collected within the Carthage-Colton Shear Zone

Stone Valley (SV): Stone Valley occurs between the village of Colton and Brown's Bridge on the Raquette River. Here, the river cuts deeply through a thick post-glacial sequence to expose the bedrock along the transition from the Highlands to Lowlands. Both ductile and brittle (Figure 3L) structural features can be readily observed [27]. An exposure of tectonic breccia consisting of large blocks (up to 3m) of lineated mylonite (Figure 3M) whose interstices are filled by a calcite–potassium feldspar–pyroxene pegmatite (Figure 3N) occurs 2 km north of the approximate trace of the CCsz as defined by mylonitic orthogneisses. This pegmatite between breccia blocks was sampled for U–Pb zircon geochronology.

About three dozen euhedral to slightly rounded zircon crystals varying in size from 100–200 μm in long dimension, were mounted. The grains were variable in their shape, zoning patterns, and BSE response. Most grains had a core and oscillatory-zoned rim (Figure 3K,L). Some cores also displayed oscillatory zoning. Some cores were darker or lighter in BSE response than the zircon that rimmed them. Forty-eight analyses were completed and yielded several concordant clusters (Figure 5L). These included concordant ages of 1041 ± 4.4 Ma (MSWD = 0.07) and 1066 ± 9.5 Ma (MSWD = 0.11). Four groups gave concordant ages of 1163 ± 2.4 Ma (*n* = 16; MSWD = 0.83); 1183 ± 3.7 Ma (*n* = 3; MSWD = 2.5); 1263 ± 22 Ma (*n* = 4; MSWD = 1.16), and 1329.3 ± 8.3 (*n* = 17; MSWD = 0.62). Respectively, they also yielded weighted averages of 1161 ± 3.8 Ma (*n* = 4; MSWD = 0.38), 1187 ± 6.7 Ma (*n* = 3; MSWD = 0.77), 1262 ± 12 Ma (*n* = 4; MSWD = 1.2), and 1332 ± 8.7 (*n* = 17; MSWD = 1.3). Notably, many of the circa 1330 Ma crystals had a 1162 Ma rim (Figure 4K). Uranium contents were moderate, ranging between 100–400 ppm. The U/Th ratio decreased from the youngest to oldest population (40.2 ± 6.3; 24.2 ± 8.3; 14.9 ± 11.1; 1.7 ± 0.6).

The crystallization age of the silicocarbonate pegmatite and brecciation event was interpreted as 1041 ± 4.4 Ma. The older ages obtained were from xenocrystic zircon many of which displayed evidence of the development of younger rims.

### 3.2. Rare-Earth Elements in Zircon

The zircon crystals analyzed showed several distinct patterns of rare-earth element enrichment (Figure 6). Samples KV, PF, SV, NB, and FN generally showed limited variation and typical patterns

associated with igneous zircon (i.e., near-chondritic LREE (Light rare-earth elements) values; positive sloping and HREE (Heavy rare-earth elements) values in the range 100–10,000× chondritic). Some zircon crystals including RR, DH, WP, HL, and JF displayed nearly flat patterns or a very slightly positive HREE slope. Three samples, ED, JF, and HL, showed a large spread of values, particularly for the LREE. Samples RR, ED, and HL were notable in having two or more types of patterns. Positive cerium anomalies were shown, to various extent, in nearly all samples investigated, but were most pronounced in KV, PF, JF (some), and NB. The development of a europium anomaly was also variable, with most negative, but a few positive (e.g., DH, WP).

Those REE patterns most closely resembling unaltered and non-metamict igneous zircon [28], had the lowest uranium concentrations and, theoretically, the least radiation damage and the least amount of inherited zircon. Those from Stone Valley (SV) showed these characteristics despite most grains being xenocrystic; however, they have low and consistent uranium concentrations. Other samples show elevated concentrations of LREE and a considerable range of values, particularly for those REE of lower atomic number. These (DH, ED, JF, HL) generally had high uranium contents, aside from the xenocryst-rich Edwards sample. Some samples showed a strong positive correlation between individual and total REE concentrations and Uranium concentrations (Figure 6J).

### 3.3. Carbon and Oxygen Isotopes in Calcite

Carbon and oxygen isotope ratios obtained from calcite collected from the samples investigated in this study yielded variable $\delta^{18}O$ (approximately 10–24‰) but more restricted $\delta^{13}C$ (approximately −3.4–0.3‰) values (Table 3). These isotope ratios ranged from values typical of mantle-derived carbonatites to values typical of carbonate sedimentary rocks (Figure 7). Whereas most replicate samples from a given location cluster tightly, one sample, FN, showed a significant range in oxygen isotope ratios among two samples ($\delta^{18}O$ 10 versus 20‰), perhaps suggesting more than one generation of calcite growth or an alteration event.

**Table 3.** Carbon and oxygen isotopes in calcite.

| Location | n | $\delta^{18}O$ | SD | $\delta^{13}C$ | SD |
|---|---|---|---|---|---|
| Edwards (ED) | 1 | 20.29 | | −0.87 | |
| Fine (FN) | 1 | 9.94 | | 0.27 | |
| Fine (FN) | 1 | 20.02 | | −0.90 | |
| Keene Valley (KV) | 1 | 11.31 | | −2.91 | |
| Powers Farm (PF) | 3 | 17.85 | 1.62 | −2.42 | 0.30 |
| Natural Bridge (NB) | 3 | 15.94 | 0.45 | −1.13 | 0.03 |
| Natural Bridge marble (1 m) | 1 | 21.38 | | 1.73 | |
| Natural Bridge marble (50 m) | 1 | 25.75 | | 1.15 | |
| Rose Road (RR) | 1 | 23.94 | | −0.27 | |
| Stone Valley (SV) | 1 | 15.20 | | −3.35 | |
| West Pierrepont (WP) | 5 | 16.99 | 1.03 | −1.74 | 0.88 |
| West Pierrepont Zn (WPz) | 1 | 11.45 | | −2.219 | |

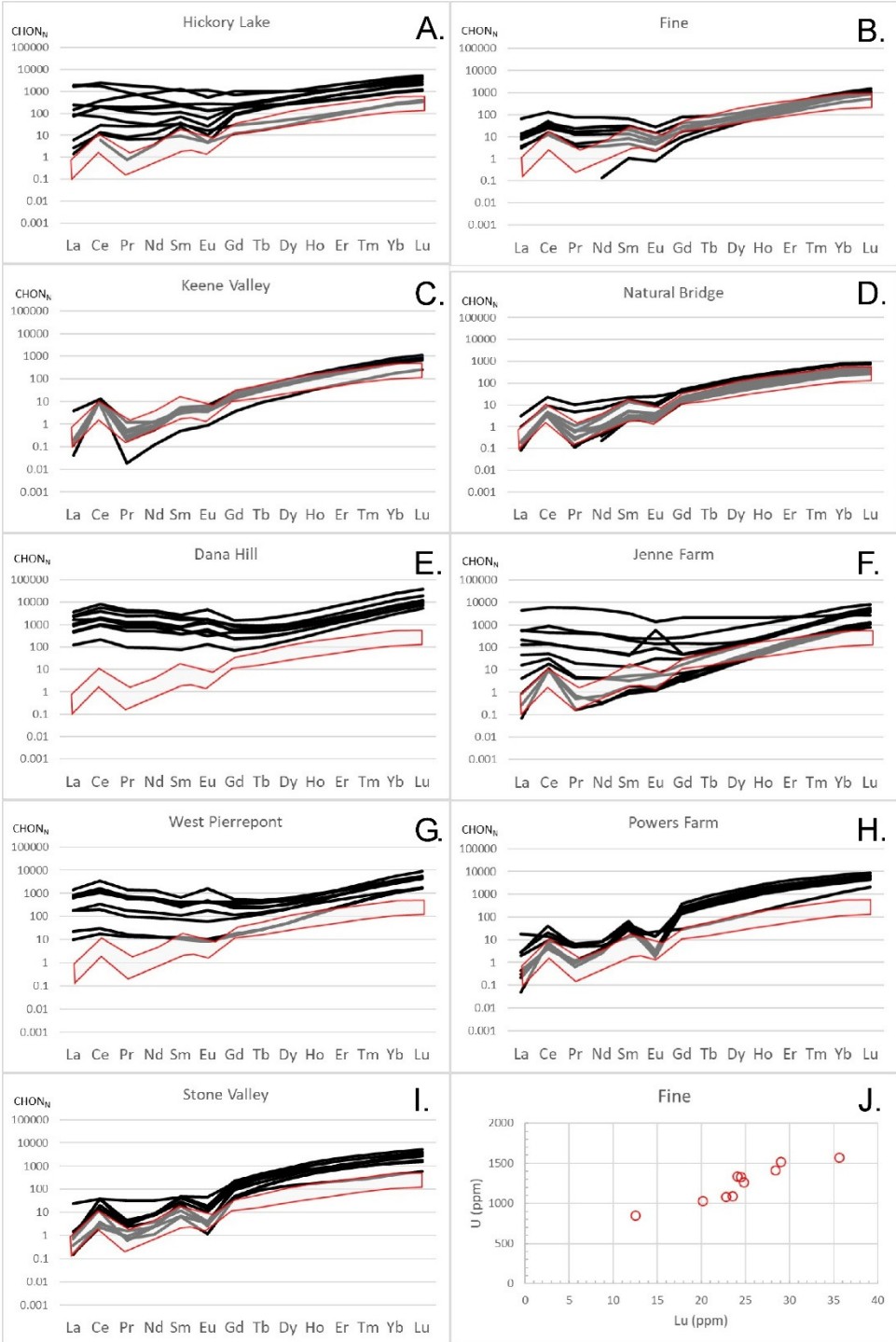

**Figure 6.** (**A–J**) Rare-earth element spidergrams for zircon samples in this study normalized to chondritic values (CHON$_N$) [29]. Continental zircon field (red outline) shown from Grimes et al. [28]. (**J**) Scatter plot of Lu versus U.

Comparisons of the silicocarbonate pegmatite samples indicate substantial overlap with marbles from the Adirondack Lowlands [30] and with calcite associated with the Balmat zinc deposits [31] indicated substantial overlap. The majority of samples fell within the field for Lowlands marble, with two samples (KV, FN) falling between metamorphosed carbonatites and marble. Similar values for $\delta^{13}$C–$\delta^{18}$O isotopes reported by other investigators [3,6,8] for calcite from geologically similar

occurrences in Canada have been interpreted as both silicocarbonatites and marble metasomatized by fluids possibly derived (at least in part) from mantle sources.

A direct comparison can be made with isotopic ratios measured by Lentz [6] for calcite from the Central Metasedimentary Belt (CMB) of Ontario. On Figure 7 they are divided into four fields including V (vein-dykes), P (associated pegmatites), S (skarns), and M (marbles), which display considerable overlap. Only the Keene Valley silicocarbonatite pegmatite from the Adirondack Highlands overlaps with the CMB vein-dykes. Nearly all the Adirondack Lowlands silicocarbonate samples fall within the skarn field.

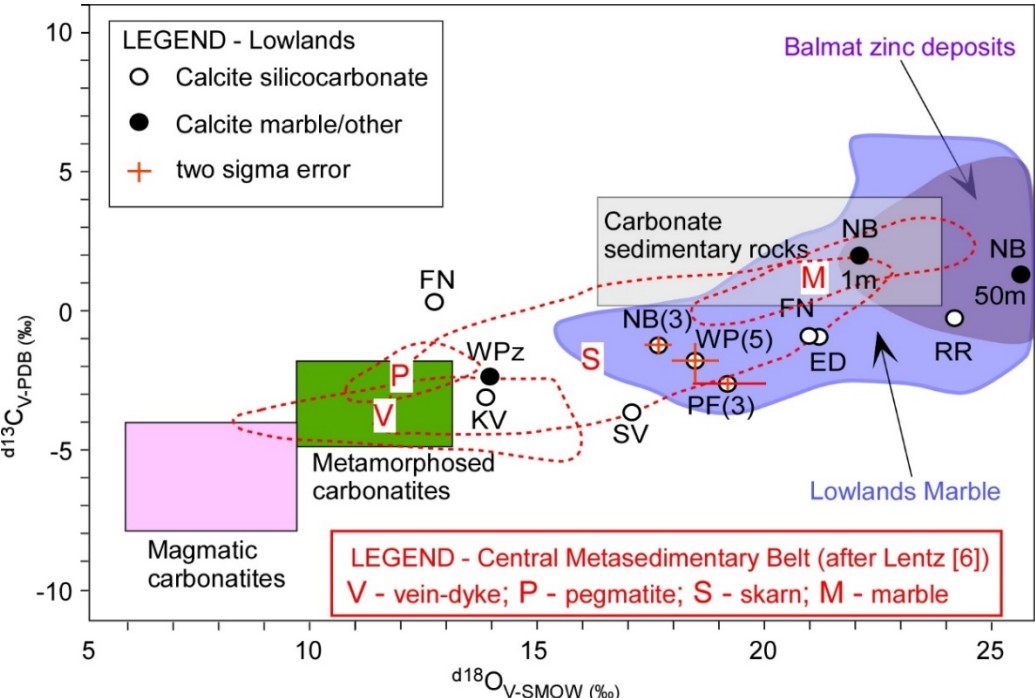

**Figure 7.** Carbon and Oxygen isotopic ratios and two sigma error bars for calcite from silicocarbonates in this study. See text for more details. Fields for magmatic carbonatites [32], for Lowlands marbles [30], for Balmat zinc deposits [31], for Natural Bridge analyses [20], and for metamorphosed carbonatites of the Central Metasedimentary Belt [3]. Fields in dashed red are from the Central Metasedimentary Belt of Ontario (Lentz [6]). Parentheses indicate the number of analyses; those without error bars were analyzed once. WPz refers to calcite associated with the zinc deposits at West Pierrepont, NY.

## 4. Discussion

### 4.1. Age of the Silicocarbonate Pegmatites

The interpretation of the ages of the silicocarbonate pegmatites investigated in this study was complicated by the occurrence of inherited zircon and Pb loss in high uranium crystals. Analyses of zircon from Edwards (ED), Rose Road (RR), and Stone Valley (SV) indicated that inherited grains made up the bulk of the zircon population, as might be anticipated from anatexis of metasedimentary sequences containing detrital zircon. A further complication was the high uranium content (>1000 ppm) and Pb-loss history observed in several other samples (Dana Hill—DH, Jenne Farm—JF, West Pierrepont—WP). Nonetheless, a number of samples had low-to-moderate uranium concentrations, minimal or no Pb loss, and few or no inherited zircon crystals, allowing well-constrained concordia ages to be calculated. In such cases, concordia ages and weighted averages are generally indistinguishable. Two samples, Natural Bridge (NB) and Powers Farm (PF), had near-concordant data clusters that fell slightly above concordia. In such cases, the weighted average was preferred and are given in Table 1 as the interpreted age. Where discordant arrays occurred, our approach was to find the zircon with the

lowest uranium concentration using BSE images as a guide and calculate upper intercept ages where possible. When upper intercept ages were used, they generally had higher error associated with them. Lower intercepts fall within the range of dozens of other samples from the Adirondacks reported in the literature (circa 200–400 Ma; e.g., [33]), adding confidence to the upper intercept age obtained.

Several geospatial trends were apparent when comparing the geochronological results across the region (Figure 8). The first was that granitic and silicocarbonates pegmatites in the Adirondack Lowlands generally yield Shawinigan crystallization ages (1175–1151 Ma), whereas those in the Highlands yield Ottawan ages (1060–1024 Ma). In contrast, in the Central Metasedimentary Belt of Ontario, Moecher et al. [3] obtained both ages (1089 ± 5, and 1143 ± 8, and 1170 ± 3 Ma) from metamorphosed carbonatites. In addition, titanite from bounding skarns yielded ages of 1035–1085 Ma. Either several events of carbonatite intrusion occurred, or existing bodies were remobilized during the Ottawan event. While the examples analyzed by Moecher et al. [3] were interpreted as metamorphosed carbonatites, vein-dykes, of greater similarity to those of the Adirondacks, occur in the Central Metasedimentary Belt but are currently undated.

In the Adirondack Region, the Carthage-Colton shear zone (CCsz) is an important boundary associated with several changes (e.g., proportions of lithologies, metamorphic grade, age of last deformational event). A second trend within the Highlands is that granitic [34] and silicocarbonate pegmatites in the eastern Adirondack Highlands appear to be 20–30 million years younger than those in the western part near the CCsz. Chiarenzelli et al. [35] noted a similar trend in the Lyon Mountain granite, suggesting that orogenic collapse may have begun in the northwest Lowlands and propagated eastward over several tens of millions of years. If so, the spatial trends in pegmatitic and leucogranitic rocks noted across the Adirondacks may be a function of time and the duration or magnitude of fault motion.

Several samples (DH, ED, JF, and SV) have an Ottawan age and are located on or just northwest of the trace of the CCsz. This is significant because of the general lack of Ottawan zircon ages from the Lowlands noted in previous studies [33]. However, the CCsz is a zone of variable width, significant sinuosity, and a northwestern dip. The silicocarbonate pegmatite from Edwards (ED) lacks Shawinigan age zircon and yields an Ottawan age but is in the Lowlands based on its position with respect to the mapped trace of the CCsz on the Adirondack Sheet of the New York State bedrock geologic map [11]. However, a field investigation indicates that the Edwards outcrop is just south (0.5 km) of a thick package of intensely strained syenitic mylonite, with intensely deformed meter-scale layers of metasedimentary rocks interpreted as inclusions of wallrock. In contrast to many of the silicocarbonate occurrences in the Lowlands, at Edwards, large clinopyroxene, rather than amphibole, is the dominant mineral. Tremolite occurs only sporadically throughout the outcrop as isolated small crystals, and along late seams or fractures. Therefore, it is possible that at the Edwards locality, either the CCsz is north of the location shown on the Adirondack Sheet or, more likely, the Edwards outcrop is within an area of greater structural complexity. The Ottawan intrusion age may indicate that melts of metasedimentary rocks technically in the Highlands terrane pierced the CCsz and were emplaced in the overlying Lowlands during late, orogenic collapse as originally hypothesized by Selleck et al. [36].

### 4.2. Origin of the Silicocarbonate Pegmatites

As noted in the introductory section, the silicocarbonate pegmatites found in the Adirondacks share many features with those found in the Central Metasedimentary Belt of Ontario and Quebec. These include the occurrence of salmon pink calcite as a matrix phase hosting large, euhedral, calc–silicate minerals, and a vein or dike-like cross-cutting relationship to older country rocks, almost exclusively those of the Grenville Supergroup. Numerous differences also occur, chiefly the major and accessory minerals found. While large or voluminous apatite (Bear Lake diggings, ON; Otter Lake, QC), fluorite (Dwyer Mine, Nu-Age Mine, ON), silica-undersaturated silicates (Princess Sodalite mine, Egan Chutes and Davis Hill/Davis Quarry, ON), betafite and pyrochlore (Silver Crater Mine, ON), uraninite (Cardiff Mine, ON; Richardson Mine, ON; Payne-Chabut prospect, QC) crystals, among others [10], are

found in several of the locations in the Central Metasedimentary Belt, they are generally not large or particularly abundant in the Adirondack examples. However, it is true that the Adirondack examples also vary widely in terms of the mineralogy. While it would be tempting to speculate that they share similar origins [8] based on their close association with Grenville Supergroup metasedimentary rocks, further study of the wide variety of Canadian examples, including their age, is warranted before convincing arguments can be made.

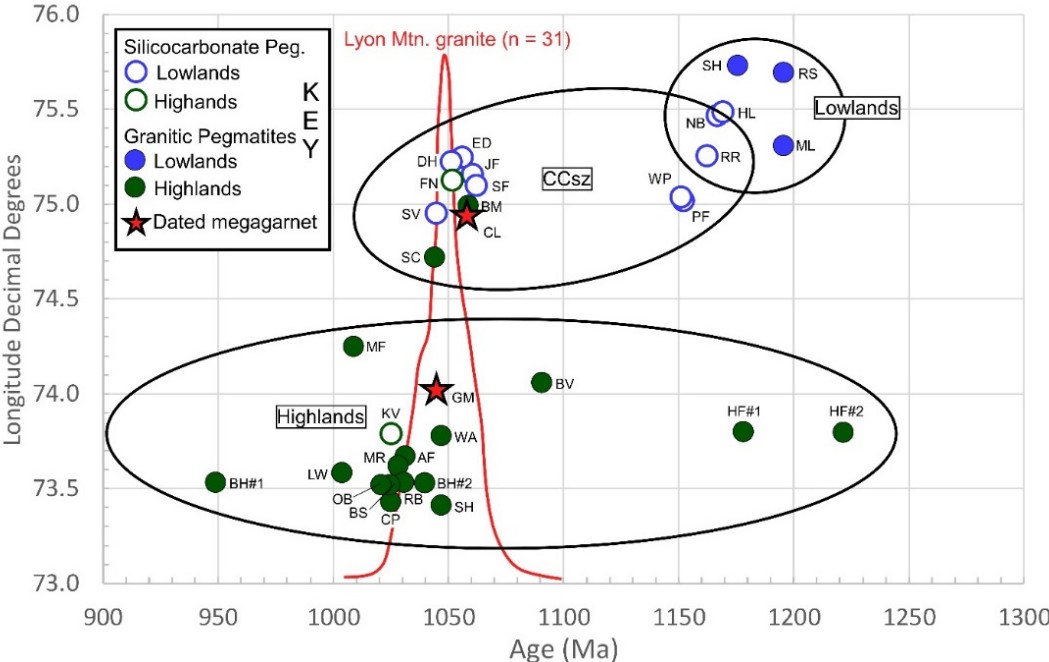

**Figure 8.** Plot of longitude versus age of pegmatites in the Adirondack Region. Green symbols are from the Highlands; blue from the Lowlands. Open circles are silicocarbonate pegmatites; closed circles are granitic pegmatites; red stars are dated megagarnet localities. The red histogram is from Reference [35] and shows the range of ages for the Lyon Mountain granite in the Highlands. Abbreviations: Barton Hill (BH #1, #2); Batchellerville (BV); Crown Point (CP); Hulls Falls (HF #1, #2); Lewis Mine (LW); McLear (ML); Mayfield (MF); Old Bed (OB); Scotts Farm (SF); Sugar Hill (SH–Highlands); Roes Spar Bed (RB); Rossie (RS); Reference [33]; Dana Hill (DH); Edwards (ED); Fine (FN); Hickory Lake (HL); Jenne Farm (JF); Keene Valley (KV); Natural Bridge (NB); Powers Farm (PF); Rose Road (RR); Stone Valley (SV); West Pierrepont (WP); Source: this study; Gore Mountain (GM); Cranberry Lake (CL); Reference [37]. Ausable Forks (AF); Benson Mines (BM); Bonanza Shaft (BS); Mineville Road (MR); South Hammond (SH–Lowlands); Warrensburg (WA); Source: authors' unpublished data.

Aside from the presence of large calc–silicate minerals in a calcite matrix, several other features appear to be common among the silicocarbonate pegmatite occurrences in this study. These include field occurrence and extent, deformational state, crystallization ages, and carbon and oxygen isotopic signature. These features suggest that these rocks crystallized from melts and, because of differences in mineralogy, the melts were also of variable composition. A conclusion reached by Schumann et al. [9], who identified numerous globular inclusions of various minerals including calcite interpreted to be fluxed silicocarbonate melt droplets in apatite in the Otter Lake "skarn" zone. A magmatic origin and knowledge of their crystallization age provide important constraints on the origin and the tectonic history of the region.

Most of the silicocarbonate pegmatite occurrences are of limited extent (generally >10,000 m$^2$) in area and crosscut host rocks that were previously deformed and metamorphosed to upper amphibolite or granulite facies. Most occurrences appear to be irregular in shape, although some such as the West Pierrepont (WP) and Dana Hill (DH) danburite, occurrences may be, at least in part, stratabound and derived, and remained, more or less in situ. At West Pierrepont, exposure of silicocarbonate pegmatites

can be traced along strike for several kilometers, linking several historic collecting localities [16,27]. In contrast, at least some collecting localities such as the Powers Farm (PF) black tourmaline locality, they are found in parallel sets of vertical dikes. The majority of the host rocks are calc–silicate gneisses or marbles of the Grenville Supergroup and, if not in direct contact with these lithologies, they are frequently exposed nearby.

Many silicocarbonate pegmatites occur within a few kilometers of the CCsz, in highly strained NE-trending lithologic belts. Some occurrences, such as those in the Stone Valley transect of the CCsz, are found between large (3 m or more) blocks of mylonitic fault rock within a tectonic megabreccia (Figure 3M), clearly requiring formation and intrusion during extension and fault movement. Nearly all known examples are located in close proximity to, or in contact with granitic pegmatites that contain large perthitic microcline, albite, quartz, and pyroxene crystals, perhaps indicative of the generation of bimodal melts or immiscible liquids. Their small volume and variable mineralogy may suggest relatively restricted areas of melting in the metasedimentary sequence which differs widely in its lithologic makeup locally due to the structural complexities.

Despite the difference in age of nearly 100 million years between Shawinigan and Ottawan examples, silicocarbonates crosscut regional deformational fabrics and metamorphism, indicating intrusion relatively late in the tectonic evolution of each orogenic event. Xenocrystic zircon common in some, but not all, of these rocks is representative of the age of local bedrock (AMCG suite: circa 1160 Ma; Hermon-Piseco suite: circa 1185 Ma; and older tonalitic gneisses: circa 1330 Ma of the eastern and southern Adirondacks) and older zircon found in detrital rocks [12]. Rare xenoliths also confirm the influence of native bedrock on the origin and sourcing of the silicocarbonates.

Studies of the carbon and oxygen isotopes of the silicocarbonates show that the calcite they contain has a variable $\delta^{18}O$ (6–20‰) but near-consistent $\delta^{13}C$ (−4–0‰) ratios. These values form a nearly horizontal trend whose left end approaches the range for calcite in carbonatites and whose right end approaches that of carbonate sedimentary rocks. The trend is also very similar to calcite from iron-rich skarns [38] and overlaps local marble values [30]. Some workers have used the calcite isotopic signature to suggest a mantle contribution to the silicocarbonate magmas [8], which our data does not rule out. Based on the data available, particularly inherited zircon xenocrysts, we suggest that the silicocarbonate occurrences in the Adirondacks represent emplacement of melts formed by localized anatexis of calc–silicate and marble-rich lithologies of the Grenville Supergroup [7].

Moecher et al. [3] used geochemistry and isotopic systematics to provide evidence of a depleted mantle source for metamorphosed carbonatites in the Central Metasedimentary Belt boundary zone. Spatially, these occurrences can be tied to the northwestern margin of a failed rift zone [39], and indicate intrusion during extension and incipient rifting. The silicocarbonate pegmatites we examined differ in several respects including their grain size, mineralogy, undeformed nature, inherited zircon populations, and relatively low proportion of calcite to calc–silicate minerals. Although they can be tied to exhumation and uplift, there is little evidence of rifting at the time of their intrusion. Therefore, it appears unlikely they are true carbonatites; however, without additional studies the role of mantle-derived components cannot be completely ruled out.

Several occurrences deserve special note as possible skarn occurrences. The strongest case can be made for the Natural Bridge (NB) and Rose Road (RR) localities, which contain wollastonite, a common contact metamorphic mineral. At these localities, marble, along the southwestern trace of the CCSz, was intruded by the Diana syenite complex circa 1165 Ma, where it broadens to 10 km or more in width. The Valentine wollastonite mine is located between the two occurrences (Figure 2) and involves similar lithologies. At the Valentine mine and elsewhere along the southwestern extent of the CCsz, there was a substantial volume of granitic to syenitic magma emplaced into marble near the end of the Shawinigan Orogeny. Further work may indicate the silicocarbonates are spatially associated with skarn zones and formed synchronously where conditions were appropriate and fluxes available to facilitate melting.

*4.3. Tectonic Implications*

Although volumetrically minor, the silicocarbonate rocks of the Adirondack Region provide important constraints on the geologic history and tectonics of the area. As intrusive rocks, they record melting of carbonate-rich source rocks of the local metasedimentary sequence and mobilization of these melts well after regional deformation. The fact that they are most numerous within a few kilometers of the Carthage-Colton shear zone indicates the control of active faulting on their emplacement and the role of uplift and associated decompressional melting. The availability of boron, chlorine, carbon dioxide, water, and other potential fluxing agents in the local metasedimentary rocks may have facilitated melting in various parts of the sequence. The occurrence in Stone Valley, near Colton, NY, unequivocally ties the emplacement of some silicocarbonate melts to brittle faulting, late in the movement history of the CCsz. The timing is synchronous with uplift, exhumation, leucogranite intrusion, and widespread fluid infiltration of the Adirondack Region [35,36,40–46].

Silicocarbonate intrusion (circa 1050 Ma) corresponds with the intrusion of the Lyon Mountain granite suite and associated granitic pegmatites [33–35,45]. It also occurred during the intrusion of the iron oxide–apatite deposits of the eastern Adirondacks and associated albitization and/or alkali-metasomatism of the Lyon Mountain granite host (35,42,45,46). In addition, this was also the time of the development of skarn-type iron deposits of the western Adirondacks and megagarnet development throughout the Highlands [37].

A major question in the Adirondacks is the extent and timing of deformation associated with the Ottawan orogenic event (circa 1090–1020). While the Lowlands are thought to be part of the orogenic lid, largely escaping Ottawan effects and lacking Ottawan intrusive rocks, the Highlands record both Shawinigan and geographically limited Ottawan melting of pelitic rocks [47,48] and, presumably, associated deformation and widespread late ferroan leucogranite intrusion [35]. Until relatively recently, the major structural grain in the Highlands was thought to be Ottawan in age. This has come into question with the recognition that the east–west 30 km wide Piseco Lake shear zone, separating the Adirondack Highlands from the Southern Adirondack Terrane, is a Shawinigan strike-slip structure [49]. The recognition of circa 1050 Ma shear zones enveloping the Marcy massif [45] has led to the conclusion that the Lowlands and Highlands did not experience compressional deformation after circa 1050 Ma. Ottawan deformation, if present, must have occurred prior to 1050 Ma, in line with estimates (circa 1090–1050 Ma) from compressional structures from the western part of the Grenville [50]. The silicocarbonate pegmatites are, thus, post-orogenic, occurring during the collapse of the orogeny along extensional faults that facilitated their movement and ultimate emplacement.

## 5. Conclusions

Mineral occurrences from the Grenville Province have been exploited, largely for collecting purposes, for nearly 200 years. They represent unusual types of silicocarbonate magmas, as shown by their intrusive relations, inherited zircon, large, euhedral crystals, and variable mineralogy, among other features. In the Adirondack Region, they fall into two distinct age groupings approximately subdivided by the Carthage-Colton shear zone, the boundary between the Adirondack Highlands and Lowlands. Those north of the boundary in the Lowlands are of late Shawinigan age (circa 1160 Ma) and are likely skarns or derived melts formed in response to the intrusion of the voluminous Diana syenite or other igneous rocks into the marble-rich metasedimentary sequence. Those within the CCsz and in the Highlands to the south, are of late Ottawan age (circa 1050 Ma) and intruded during a time of orogenic collapse, extensional faulting, hydrothermal fluid flow, granite and granitic pegmatite intrusion, and formation of iron oxide-apatite and garnet ores in the Highlands. They are linked to local bedrock by their composition, trace minerals, association with calc–silicate and marble lithologies, and population of inherited zircon. An origin involving partial melting of metasedimentary bedrock during widespread anatexis driven by derived fluids and volatiles during rapid uplift and exhumation is envisioned.

**Supplementary Materials:** The following are available online at http://www.mdpi.com/2075-163X/9/9/508/s1, Supplemental File S1: U–Pb zircon geochronology of silicocarbonate pegmatite samples. Containing all U–Pb zircon data.

**Author Contributions:** This paper was primarily written by J.C. with contributions and review by the co-authors. The samples were collected by M.L., J.C., G.R., and D.B.; M.L. and J.C. analyzed the zircon samples at the Arizona Laserchron Center. J.S. analyzed rare-earth elements in zircon at Rensselaer Polytechnic Institute. D.B. assisted with the preparation of carbon and oxygen isotopes samples. G.R. provided geological background information on the mineral locations sampled. All authors contributed to the evaluation and discussion of the data.

**Funding:** This research was funded by the New York State Museum and the Archie F. and Barbara Torrey MacAllaster North Country Professorship.

**Acknowledgments:** We thank the New York State Museum (ML) and the Archie F. and Barbara Torrey MacAllaster North Country Professorship (JC) for funding. The authors thank Mark Pecha and staff at the Arizona Laserchron Center for their help with sample preparation and analysis. The NSF award EAR 1649254 to the Arizona Laserchron Center is acknowledged. Steve Chamberlain is recognized for his enthusiasm and knowledge of local mineral occurrences. Two anonymous reviewers and William Peck provided substantial suggestions for improving the paper and we gratefully acknowledge their efforts.

**Conflicts of Interest:** The authors declare no conflict of interest.

## Appendix A  Analytical Procedures

Geochronology: U–Pb zircon geochronology was carried out at the Arizona Laserchron Center, University of Arizona. Zircon crystals are shown in Figure 4. The U–Pb analytical results are shown in Figure 5 and in summary and detail in Tables 1 and 2. Supplemental File S1 contains the U–Pb zircon data. Analyses were conducted by Laser Ablation-Multi-Collector-Inductively Coupled-Mass Spectrometry (LA-MC-ICP-MS) as previously described [12,14,35,51]. Ablation of zircon grains embedded in epoxy and sectioned 1/3 of the distance through was done with a Photon Machines Analyte G2 DUV193 Excimer laser (Isomass Scientific Inc., Calgary, AB, Canada) using spot diameters of 30 μm with a fluence of ~4 J/cm$^2$ and frequency of 4 Hz. The ablated material was carried with helium–argon gas (flow rate 0.2–0.36 L/min) into the plasma source of a Nu Instruments HR ICPMS (LA-MC-ICP-MS, Nu Instruments Ltd., Wrexham, UK) equipped with a flight tube of sufficient width that U, Th, and Pb isotopes can be measured simultaneously. All measurements were made in a static mode using Faraday detectors for $^{238}$U, $^{232}$Th, and $^{208-206}$Pb, and discrete dynode ion-counters for $^{204}$Pb and $^{202}$Hg. Ion yields were typically ~1 mV per ppm. Each analysis consisted of one 15 s integration on peaks with the laser off (for backgrounds), fifteen 1 s integrations with the laser firing, and a 30 s delay to purge the previous sample and prepare for the next analysis. The ablation pit was ~4–15 μm in depth.

For each analysis, the errors in determining $^{206}$Pb/$^{238}$U and $^{206}$Pb/$^{204}$Pb resulted in a measurement error of ~1% (at 2$\sigma$ level) in the $^{206}$Pb/$^{238}$U age. The errors in measurement of $^{206}$Pb/$^{207}$Pb and $^{206}$Pb/$^{204}$Pb also result in ~1% (2$\sigma$) uncertainty in age for grains that were >1.0 Ga, but they were substantially larger for younger grains due to the low intensity of the $^{207}$Pb signal. For most analyses, the crossover in precision of $^{206}$Pb/$^{238}$U and $^{206}$Pb/$^{207}$Pb ages occurred at circa 1.0 Ga. The $^{206}$Pb/$^{238}$U ages were reported in this paper. Common Pb correction was accomplished using the measured $^{204}$Pb and assuming an initial Pb composition [52], with uncertainties of 1.0 for $^{206}$Pb/$^{204}$Pb and 0.3 for $^{207}$Pb/$^{204}$Pb. The measurement of $^{204}$Pb was unaffected by the presence of $^{204}$Hg because backgrounds were measured on peaks (thereby subtracting any background $^{204}$Hg and $^{204}$Pb) and because very little Hg was present in the argon gas. Interelement fractionation of Pb/U was generally ~20%, whereas apparent fractionation of Pb isotopes was generally <2%. In-run analysis of fragments of a large Sri Lankan zircon crystal (generally every fifth measurement through the course of analyses and multiple times at the beginning and end of each run) with a known age of 564 ± 4 Ma (2$\sigma$ error) was used to correct for this fractionation. The uncertainty resulting from the calibration correction was generally ~1% (2$\sigma$) for both $^{206}$Pb/$^{207}$Pb and $^{206}$Pb/$^{238}$U ages. Uncertainties shown in these tables were at the 1$\sigma$ level and included only measurement errors. The reported ages were calculated using Isoplot 4.0 software) [53].

Final weighted average and concordia age diagrams (Figure 5) showed two sigma uncertainties. The smaller uncertainty (labeled Mean) was based on the scatter and precision of the set of $^{206}$Pb/$^{238}$U or $^{206}$Pb/$^{207}$Pb ages, which were weighted according to their measurement errors (shown at $2\sigma$). The larger uncertainty (final age), which was the reported uncertainty of the age, was determined as the quadratic sum of the weighted mean error plus the total systematic error for the set of analyses. The systematic error, which included contributions from the standard calibration, age of the calibration standard, composition of common Pb, and U decay constants, was generally ~1% to 2% ($2\sigma$).

Analyses were conducted on zircon grains mounted in epoxy plugs and polished to yield a significant cross-section through the grain. As noted below, the grains were analyzed in a back scattered electron (BSE) or cathodoluminescence (CL) modes (or both) to identify internal features to be targeted or avoided (Figure 4). The sampling strategy involved analyzing 35 targeted areas selected prior to the analysis to check for variability that might arise from zoning, cores, and rims. Numerous grains were analyzed two or more times to check for consistency or to characterize "cores" and "rims." Cracks, fractures, inclusions, or other heterogeneities were avoided.

Trace-elements: Trace-elements in zircon were measured by LA-ICP-MS on a Photon Machines Analyte193 G1 short-pulse excimer laser coupled to Varian 820 quadrupole inductively coupled plasma mass spectrometer at RPI (Isomass Scientific, Inc., Calgary, AB, Canada). All trace-elements were calibrated using NIST (National Institute of Standards and Technology, Washington, WA, USA) standard glass #610. Laser parameters included 360 shots at 6 Hz repetition rate per ablation at 58% laser power attenuation and 6.5 J/cm$^2$ fluence using 20 μm square spot. Mass spectrometer acquisitions were 140 s windows each including 60 s ablation, 20 s washout, and 60 s background acquisition. Masses, analyzed by LA-ICP-MS (with millisecond dwell times in parentheses) included $^{43}$Ca (30), $^{45}$Sc (10), $^{49}$Ti (1), $^{55}$Mn (0.2), $^{57}$Fe (1), $^{89}$Y (10), $^{90}$Zr (10), $^{118}$Sn (10), $^{139}$La (30), $^{140}$Ce (30), $^{141}$Pr (30), $^{146}$Nd (30), $^{149}$Sm (20), $^{153}$Eu (20), $^{157}$Gd (20), $^{159}$Tb (20), $^{163}$Dy (10), $^{165}$Ho (10), $^{166}$Er (10), $^{169}$Tm (10), $^{172}$Yb (10), $^{175}$Lu (10), $^{178}$Hf (10), $^{182}$W (10), $^{206}$Pb (10), $^{232}$Th (20), and $^{238}$U (30). The total quadrupole scan time was 465 ms per cycle, which resulted in >120 cycles within each 60 s ablation event. Each zircon sample was ablated five times in distinct locations, and standards were reanalyzed every five ablations. Data were processed using the Iolite software's trace-element data reduction scheme using Si as an internal standard.

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
