# Peer review of "Age and Origin of Silicocarbonate Pegmatites of the Adirondack Region"

_minerals, doi:10.3390/min9090508_

Round 1
Reviewer 1 Report
Overall, I am happy with the description of the occurrences, as well as the geochronology and light stable isotope work reported in this paper. There is little or no need for improvement with any of these. The methods, analyses and the conclusions drawn are sound. However, I am struggling with the silicocarbonate melt interpretation for the pegmatites (lines 84, 557, 577, 584, 587, 608). The authors cite numerous examples of past interpretations of the origin of these unusual rocks, and they are about as varied as possible. It appears that the silicocarbonate melt interpretation is supported largely from inferred melt inclusions described in apatite from a 2017 abstract authored by Shulman et al. Although I am a strong supporter of the use of crystallized melt inclusions as evidence for magmatic origin, I worry about this interpretation because it contrasts with loads of experimental work that show magmatic immiscibility between silica-rich and carbonate-rich melts. Consequently, do the authors infer that the Ottawan-age pegmatites intruded as miscible silicocarbonate melts? Or did they intrude as an immiscible melts? Could two melts have intruded at different times? I think the first question is reasonable. Can the authors address this?
The rest of my comments are minor or editorial in nature. I would ask the authors to include a general statement on where they believe the analyzed zircon grains are found . . . as inclusions in each of the silicate phases? As inclusions in the host calcite? Both? I did not rigorously check the references.
Figure 2: The Bpg unit is uncolored in the legend, but very pale yellow on the map; greenish-brown map areas occur north and southwest of Gouverneur, but do not appear in the legend; I believe there are anorthosite suite rocks to the northeast of Carthage that don’t show on the map; there are two pink-colored map areas northeast of Hermon and one east of Gouverneur that are either not in the legend or are miss-colored leucogranites; there is no contact between Q and light green granitic/syenitic rocks just southwest of Parishville; the amphibolite just to the southwest of Pierrepont looks to be dark gray rather than black.
Lines 183, 248: I didn’t know there was red corundum at Rose Rd, that's unusual and important.
Lines 187, 289: This is minor, but has the chalcopyrite identification been confirmed? I always thought it was just pyrite with an iridescent coating. I know the molybenite has been confirmed.
Line 191: maybe this could be better described as “Tarp-covered trench at the Powers Farm.” so it doesn’t seem as if you are describing a picture of a tarp in your research paper!
Line 205: by diameter, do you mean length or width?
Lines 317, 353, 424: uranium rather than Uranium.
Line 336: delete “locality”
Reviewer 2 Report
This is an important paper with new age dating of some unusual pegmatites from the Mesoproterozoic Adirondack Mountains. Most of these localities are well-known mineral collecting localities visited by amateurs and mineral clubs, and have yielded many museum specimens. These rocks have been seen by a lot of geologists on field trips, but until now they have not been tied well to our understanding of Adirondack geologic history, or looked at with a modern petrologic eye. The new age data helps constrain their genesis both in age and source materials, and is a welcome addition to Adirondack and Grenville geology.
As said by the authors, these rocks related to a well-known group of similar rocks– the vein-dikes of the Canadian Grenville Province. These rocks often contain orange or salmon calcite and large apatite crystals. I think it is worth adding a bit more discussion to this connection, perhaps by listing similarities and differences (are they orange? how common is apatite?, etc.). I think it would also be good to mention a few of the key localities, like the Bear Lake Diggings, the Silver Crater Mine, and Egan Chute (Otter Lake is already named)– people are likely to find your paper by google searching, and having these in the text will help folks find it.
On this topic I think it is worth mentioning that the Moecher localities are different from many of the more famous fluorite+apatite+calcite vein-dikes. I have worked on both, and the Moecher localities are just more convincing as meta-carbonatites: they have more igneous chemistries and contain a lot of disseminated apatite and micas, and are loaded with zircon that range from silt-sized to pegmatitic. The O and C isotopes of these are closer to igneous values than the fluorite+apatite+calcite vein-dikes too. I am glad that you showed a field for the Moecher carbonatites on Fig. 7, and I would argue that you should add a field for the Canadian fluorite+apatite+calcite vein-dikes as well, as it would overlap a lot of your data. A good field can be traced from Lentz (1998) Fig 36, or a box from their averages on that page.
In some ways the O and C isotopes for these rocks are equivocal. They certainly have a component that is similar to marble country rocks, but there could be many meanings of the lower d18O and d13C values: they could represent a carbonatite component, an igneous fluid component, or a fluid that has interacted with so many regional metaigneous rocks that it has taken on the lower d18O and d13C. CCSZ veins have very similar O and C isotopes as your rocks: https://gsa.confex.com/gsa/2008NE/finalprogram/abstract_135229.htm
I agree that skarns often have these kinds of trends (line 554), and it is very common; I would cite a more general compilation paper for this like Baumgartner and Valley (RIMG 43, 2001) or Bowman (MAC Short Course 26, 1998).
On p. 19 the authors have an interesting observation, about Ottawan ages in the Lowlands, and propose the idea that these could come from the underlying Lowlands. This is similar to something that Bruce Selleck hypothesized, based on Ottawan titanite ages in a vein on the Lowlands side of the CCSZ (abstract referenced above).
In the descriptive text there are some problems with the use of past vs present tense. I think this comes from the transition of the past tense describing analytical details to the description of the results, which should be in the present tense. Most of the text does use the present tense for descriptions of zircon chemistry, zoning, or shapes, but sometimes the past is used. For example, line 227 should read “Uranium content is low”, not “was low”. Places where I noticed this are lines 227, 242, 243, 292, 332, 353, 370, 392, 393, 394 (2x), + 395.
Other comments
Line: comment or suggestion
18: superscript ‘2’
42: change to ‘well-known’
78: I would say ‘carbon and oxygen isotope ratios’
101: Is the Selleck Rd locality here they same as the WP locality? Either way it should be mentioned for a reader who is trying to make sense of descriptions of these places
124: I know this might be a hassle, but it would be great to have the ‘primary reference’ for each locality in this table
143: It is worth saying that phosphoric acid was used, the temperature, and the acid fractionation factor here.
144: Are there international standards analyzed as unknowns, or can you report lab values for them? Is there a reference for the lab’s technique or standards?
151: should read ‘summarize’
152: missing “U” in “U-Pb”
161 and elsewhere: I would either use ‘back-scattered’ or ‘backscattered’, but ‘back scattered’ doesn’t seem right to me.
262 and 263: missing ‘Ma’
266: I think this should be ‘xenocrystic’
317 and elsewhere: Element names should only be capitalized at the beginning of a sentence
364: I printed out the article and had trouble reading the inset box text on each part of this figure
369: delete the comma
376: ‘Fig. 5K’
376: I would spell out uranium to begin a sentence.
387: I really like this summary table– it is a nice, traceable explanation of the many choices that needed to be made to calculate the ages.
427: ‘near consistent’ is awkward-sounding to me. How about ‘constrained’?
428: I would avoid describing the diagram, and would rather hear a description of the data. This is partly because terms like ‘horizontal’ depend on which isotope ratio is plotted on which axis. Could this read something like “These isotope ratios range from those of carbonatites to…”?
440: Ditto the earlier comment for the use of ‘above’ here.
473: comma after ‘(1175-1151 Ma)’
481: no comma needed after ‘Highlands’
500: specify that this is the NYSM map?
529: This is a bit awkward. Also, I was wondering what the evidence was for in-situ derivation, vs concordant injection into a fabric?
550 paragraph: this repeats the earlier description of the data too closely. I would change phrases ‘near consistent’ and ‘horizontal trend’
572: no comma needed after ‘mine’
573: comma after ‘(Fig. 2)’
573: ‘There is a’
633: SF should be ‘NSF’? I don’t think it’s necessary; in the award number.
637: Isotopes need superscripts this section
Reviewer 3 Report
My comments are in the attached document.
